# OPENTSLM: TIME-SERIES LANGUAGE MODELS FOR REASONING OVER MULTIVARIATE MEDICAL TEXT- AND TIME-SERIES DATA

## ABSTRACT

Large language models (LLMs) have emerged as powerful tools for interpreting multimodal data (e.g., images, audio, text), often surpassing specialized models. In medicine, they hold particular promise for synthesizing large volumes of clinical information into actionable insights and patient-facing digital health applications. Yet, a major limitation remains their inability to handle time series data. To overcome this gap, we present OpenTSLM, a family of Time-Series Language Models (TSLMs) created by integrating time series as a native modality to pretrained LLMs, enabling natural-language prompting and reasoning over multiple time series of any length. We investigate two architectures that differ in how they model time series. The first, OpenTSLM-SoftPrompt, models time series implicitly by concatenating learnable time series tokens with text tokens via soft prompting. Although parameter-efficient, we hypothesize that explicit time series modeling scales better and outperforms implicit approaches. We thus introduce OpenTSLM-Flamingo, which integrates time series with text via cross-attention. We benchmark both variants with LLaMa and Gemma backbones against baselines that treat time series as text tokens or plots, across a suite of text–time-series reasoning tasks. We introduce three time-series Chain-of-Thought (CoT) datasets: HAR-CoT (human activity recognition), Sleep-CoT (sleep staging), and ECG-QA-CoT (ECG question answering). Across all, OpenTSLM models consistently outperform baselines, reaching 69.9% F1 in sleep staging and 65.4% in HAR, compared to 9.05% and 52.2% for finetuned text-only models. Notably, even 1B-parameter OpenTSLM models surpass GPT-4o (15.47% and 2.95%). OpenTSLM-Flamingo matches OpenTSLM-SoftPrompt in performance and outperforms on longer sequences, while maintaining stable memory requirements. By contrast, SoftPrompt exhibits exponential memory growth with sequence length, requiring 110 GB compared to 40 GB VRAM when training on ECG-QA with LLaMA-3B. Expert reviews by clinicians find strong reasoning capabilities and temporal understanding of raw sensor data exhibited by OpenTSLMs on ECG-QA. To facilitate further research, we provide all code, datasets, and models as open-source resources.

## 1 INTRODUCTION

Medicine is inherently temporal: assessment, diagnosis, and treatment depend on how signs, symptoms, and biomarkers evolve over time Giannoula et al. (2018); Henly et al. (2011); Jørgensen et al. (2024). Clinical decision-making relies on temporal patterns—tracking vital signs, medication responses, laboratory values, and disease progression markers to guide diagnosis, prognosis, and therapeutic interventions. As time-series data from electronic health records and continuous monitoring proliferate Abernethy et al. (2022); Marra et al. (2024); Yeung et al. (2023), human-legible representations become essential for interpreting and managing this information Olex & Mcinnes (2021); Senathirajah et al. (2020); Zhou et al. (2008). Clinical summaries must translate complex temporal patterns—hemodynamic instability, biomarker trajectories, and treatment responses—into interpretable assessments that support evidence-based decision-making and care coordination.

Recent advances in multimodal large language models (LLMs) allow users to interpret complex data through natural language, synthesizing information across text, images, audio, and video Wu et al. (2023); AlSaad et al. (2024). However, reasoning over longitudinal time series data remains a critical blind spot among currently supported modalities. Prior work has attempted to integrate

time-series as plain text tokens Gruver et al. (2023); Kim et al. (2024); Liu et al. (2023); however results have been limited Merrill et al. (2024). Other approaches reprogram LLMs to act as feature extractors for classification heads, which then output a fixed set of classes or values, thereby losing text-generation capabilities Li et al. (2025); Nie et al. (2023); Pillai et al. (2025); Ye et al. (2025). More recently, soft prompting has been explored, concatenating learnable time-series tokens with text tokens to preserve generation Chow et al. (2024). Yet, longer series may require more tokens, increasing context length Götz et al. (2025); Nie et al. (2023) and compute due to the quadratic cost of self-attention Nie et al. (2023); Vaswani et al. (2017).

To overcome prior limitations, we propose Time-Series Language Models (TSLMs), which integrate time series as a native modality in LLMs. TSLMs provide a natural interface to complex medical data, enabling clinicians and patients to query, interpret, and reason about longitudinal health information directly through natural language. We introduce OpenTSLM, a family of TSLMs built by extending pretrained LLMs with time-series inputs. A central design question in building TSLMs is how to represent time-series signals. Prior work has primarily used soft prompting, encoding time series as learned token embeddings concatenated with text tokens. While lightweight, this captures temporal dependencies only implicitly, as additional tokens in the context, and may scale poorly to longer or multiple sequences. We hypothesize that explicit multimodal fusion via cross-attention may be more effective for modeling temporal structure. To compare both approaches, we explore two variants for OpenTSLM. The first, **OpenTSLM-SoftPrompt**, models time series implicitly by encoding the time series into tokens and concatenating them with text tokens via soft prompting, so the model processes both as a single sequence without distinguishing between them. The second, **OpenTSLM-Flamingo**, by contrast, models time series explicitly as a separate modality, using a cross-attention mechanism inspired by Flamingo Alayrac et al. (2022) to fuse time-series and text. We created OpenTSLM-SoftPrompt and OpenTSLM-Flamingo using Llama Touvron et al. (2023) and Gemma GemmaTeam et al. (2024) backbones. We benchmark these models against each other and against baselines including LLMs with tokenized time-series inputs Gruver et al. (2023), fine-tuned tokenized time-series models, and vision-based approaches. Unlike prior classification-based approaches, our models are trained in text-based reasoning tasks, generating chain of thought (CoT) rationales before producing predictions. For training and evaluation, we introduce three new datasets: **HAR-CoT**, **Sleep-CoT**, and **ECG-QA-CoT**. To foster reproducibility and further research on TSLMs, we release OpenTSLM as an open-source framework, including models and datasets[1].

## 2 RELATED WORK

Creating Time-Series Language Models remains an open research challenge. The main barrier is the modality gap between continuous signals and discrete text representations Chow et al. (2024); Pillai et al. (2025); Zhang et al. (2025). Prior work has proposed three main strategies to bridge this gap, as summarized by Zhang et al. (2024): tokenizing time series as text (Section 2.1), applying soft prompting (Section 2.2), and using cross-attention mechanisms (Section 2.3). Table 1 provides an overview of relevant methods.

---

[1]https://github.com/StanfordBDHG/OpenTSLM

Table 1: Methods combining time-series data with LLMs.

| Name | Method | Task | Text Gen. | Multi-Sensor | Raw Data | SFT |
|---|---|---|---|---|---|---|
| FSHLLiu et al. (2023) | Token | CL | ✔ | ✔ | ✔ | |
| Gruver et al. (2023) | Token | FC | ✔ | | ✔ | |
| HealthLLM Kim et al. (2024) | Token | TR | ✔ | ✔ | ✔ | ✔ |
| Chow et al. (2024) | Soft Prom. | TR | ✔ | ✔ | ✔ | ✔ |
| ChatTS Xie et al. (2025) | Soft Prom. | TR | ✔ | ✔ | ✔ | ✔ |
| ITFormer Wang et al. (2025) | Soft Prom. | TR | ✔ | ✔ | ✔ | ✔ |
| InstrucTime Cheng et al. (2025) | Soft Prom. | TR | ✔ | ✔ | ✔ | ✔ |
| MedTsLLM Chan et al. (2024) | Soft Prom. | CL | | ✔ | ✔ | |
| MedualTime Ye et al. (2025) | Soft Prom. | CL | | | ✔ | ✔ |
| SensorLLM Li et al. (2025) | Soft Prom. | CL | | ✔ | ✔ | ✔ |
| Time2Lang Pillai et al. (2025) | Soft Prom. | CL | | | ✔ | |
| **OpenTSLM-SP (ours)** | Soft Prom. | TR | ✔ | ✔ | ✔ | ✔ |
| SensorLM Zhang et al. (2025) | Cross.Attn. | CL | ✔ | ✔ | | |
| **OpenTSLM-Flamingo (ours)** | Cross.Attn. | TR | ✔ | ✔ | ✔ | ✔ |

CL =Classification, FC =Forecasting, TR =Text Reasoning

## 2.1 TOKENIZATION OF TIME SERIES AS TEXT INPUTS

Gruver et al. has demonstrated that LLMs can perform time series forecasting by encoding values as text tokens and predicting future values without domain-specific tuning Gruver et al. (2023).Liu et al. (2023) tokenize data from wearables and smartphones to enable LLMs to infer clinical and wellness information through few-shot prompting. Similarly, Kim et al. (2024) propose HealthLLM, a framework for health prediction using physiological signals (e.g., heart rate, sleep) combined with user context and medical knowledge embedded in prompts.

## 2.2 COMBINING TEXT AND TIME SERIES TOKEN EMBEDDINGS (SOFT PROMPTING)

An alternative to manual tokenization is to encode time series into embeddings that capture time series information, using a time series encoder as presented by Nie et al. (2023). These embeddings can be input into a transformer directly or concatenated with text embeddings (softprompting) Chow et al. (2024); Nie et al. (2023); Pillai et al. (2025); Ye et al. (2025); Xie et al. (2025); Wang et al. (2025); Cheng et al. (2025); Chan et al. (2024). Pillai et al. (2025) use this approach and train an encoder to produce soft prompts from time series, which are then processed by a frozen LLM for classification via a projection head; however, this disables free-form text generation. Ye et al. (2025) and Chan et al. (2024) similarly combine time series and text-token embeddings, using a classification head Ye et al. (2025) and a task solver Chan et al. (2024) for prediction. Wang et al. (2025) introduced ITFormer, a novel framework that combines any time-series encoder with any frozen LLM to support time series question answering, also by combining text- and derived time-series tokens. Cheng et al. (2025) introduce a framework that first aligns time-series and natural language in a general stage, and later finetunes for a specific domain to perform classification. Li et al. (2025) integrate sensor and text embeddings in two stages: First, they generate a caption-like summary of the time series for free-form output; Second, they classify the data via a projection head, therefore restricting free from output. Chow et al. (2024) and Xie et al. (2025) interleave time series tokens with text tokens in the LLM input, enabling free-form text reasoning.

## 2.3 CROSS-ATTENTION FOR TIME-SERIES DATA

Few studies use cross-attention to integrate time series into LLMs. Zhang et al. (2025) apply cross-attention between a time series encoder and a text encoder, aligned with contrastive loss, to extract statistical summaries (e.g., mean, max) from a single sensor. They train a new sensor encoder, text encoder, and multimodal text decoder, rather than adapting a pretrained LLM Zhang et al. (2025).

## 3 METHODS

We present two architectures for TSLMs, OpenTSLM-Soft Prompting (SP) (Section 3.2 and OpenTSLM-Flamingo (Section 3.3). To support multiple time-series inputs, we design a prompt format that interleaves sensor data with accompanying textual descriptions (e.g., "Data from Sensor X over Y days:" followed by the data representation). Figure 1 illustrates our approach.

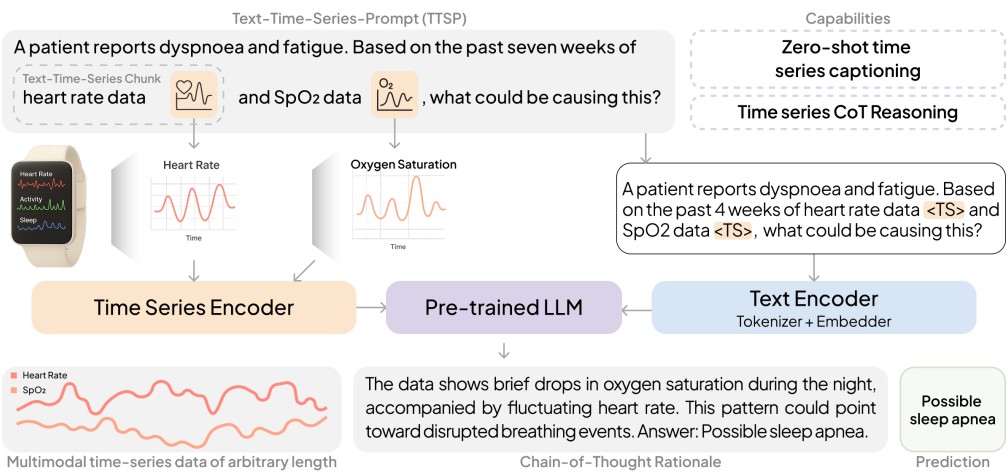

Figure 1: Overview of Text–Time-Series LLMs with support for multiple time-series inputs.

## 3.1 TIME-SERIES ENCODER

Both OpenTSLM architectures use a time series encoder inspired by Nie et al. (2023). It consists of a `Patchencoder`, followed by either a `TransformerEncoder` for OpenTSLM-SP or a `PerceiverResampler` for OpenTSLM-Flamingo (inspired by Alayrac et al. (2022); Awadalla et al. (2023)). We divide an input time series $x \in \mathbb{R}^L$ into non-overlapping patches of size $p$, yielding $N = L/p$ patches. Each patch is then transformed into an embedding vector using a 1D convolution and added with a positional encoding Nie et al. (2023)

$$\text{Patch Embedding: } \mathbf{E}_i = \text{Conv1D}(x_{i \cdot p:(i+1) \cdot p}) \in \mathbb{R}^{d_{\text{enc}}} + \mathbf{P}_i \tag{1}$$

where the convolution has kernel size and stride equal to $p$, mapping each patch to a $d_{\text{enc}}$-dimensional embedding. $\mathbf{P}_i$ is the learnable positional encoding. The sequence of position-augmented embeddings is then processed by the specific Encoder (cf. Sections 3.2 and 3.3).

**Preserving scale and temporal information** The `PatchEncoder` expects inputs normalized to $x \in [-1, 1]$. Since raw time series differ in scale and resolution across modalities depending on the sensor. Consistent with prior work Chow et al. (2024); Xie et al. (2025) we preserve scale and temporal context by adding the original mean, standard deviation, and time scale to the textual description. For example:

*This is heart-rate data over **24 hours sampled at 50 Hz** with **mean=61** and **std=12***.

## 3.2 SOFT-PROMPTING ARCHITECTURE (OPENTSLM-SP)

OpenTSLM-SP has three components: (1) a time series encoder that transforms raw data into patch embeddings, (2) a projection layer mapping embeddings to the LLM hidden space, (3) a pretrained LLM, fine-tuned using LoRA adapters Hu et al. (2021) Figure 2 illustrates the architecture.

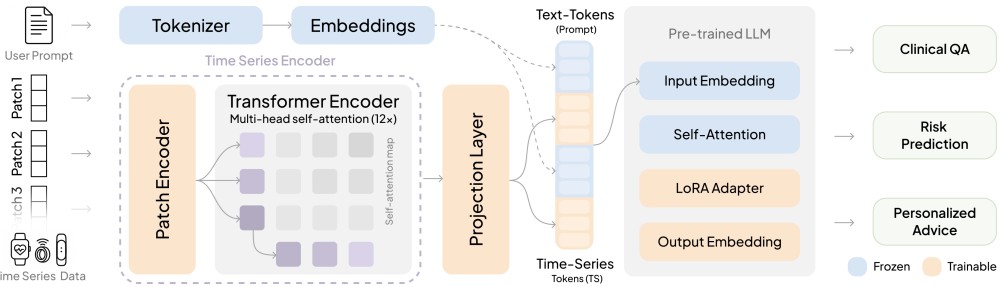

Figure 2: Architecture of OpenTSLM-SoftPrompt

**Projecting Time-Series Tokens to Text Tokens**   We apply the patch embeddings to a transformer encoder and subsequently project the resulting tokens with an multi-layer perceptron (MLP) to align them with the embedding space of dimension $d_{\text{llm}}$, corresponding to the hidden size of the language model, following Nie et al. (2023) and Chow et al. (2024).

$$\mathbf{Z} = \text{MLP}(\textbf{TransformerEncoder}(E_{1:N})) \in \mathbb{R}^{N \times d_{\text{llm}}} \tag{2}$$

where $\mathbf{Z} \in \mathbb{R}^{N \times d_{\text{llm}}}$ denotes the projected time-series tokens in the LLM embedding space.

**Text-Time-Series integration via Soft Prompting**   We interleave any number of text and time-series tokens through a soft prompting mechanism. A typical prompt consists of (1) an initial text segment ("pre-prompt"), (2) a sequence of interleaved time-series tokens and textual descriptions, and (3) a final text segment ("post-prompt"), often a question. Formally, the model input is:

$$\mathbf{X}_{\text{input}} = [\mathbf{T}_{\text{pre}}, \mathbf{Z}_1, \mathbf{T}_{\text{desc}_1}, \mathbf{Z}_2, \mathbf{T}_{\text{desc}_2}, \ldots, \mathbf{Z}_K, \mathbf{T}_{\text{desc}_K}, \mathbf{T}_{\text{post}}] \tag{3}$$

where $\mathbf{T}_{\text{pre}}$, $\mathbf{T}_{\text{desc}_i}$, and $\mathbf{T}_{\text{post}}$ are token embeddings of text segments, and each $\mathbf{Z}_i$ is a projected time-series embedding aligned with the LLM hidden space. We refer to each $(\mathbf{Z}_i, \mathbf{T}_{\text{desc}_i})$ as a text–time-series chunk. This approach implicitly integrates time series through learned tokens.

### 3.3   Cross-attention architecture (OpenTSLM-Flamingo)

OpenTSLM-Flamingo is inspired by the Flamingo model for vision–language tasks Alayrac et al. (2022); Awadalla et al. (2023). Following OpenFlamingo Awadalla et al. (2023), we extend pre-trained LLMs with cross-attention layers to support time-series reasoning.

**Architecture Overview**   We replace the vision encoder of Flamingo with a time series encoder and adapt the cross-attention mechanism for temporal data. The model consists of: (1) a time series patch encoder, (2) a Perceiver Resampler, (3) gated cross-attention layers integrated into the LLM, and (4) the frozen language model backbone. Figure 3 visualizes the architecture.

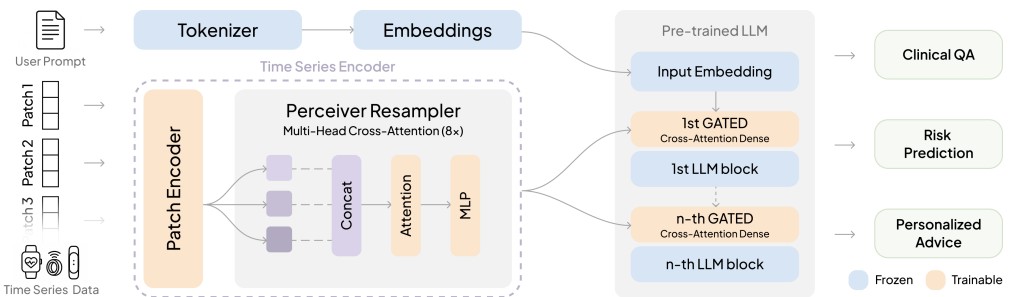

Figure 3: Architecture of OpenTSLM-Flamingo

**PerceiverResampler**   We use a PerceiverResampler inspired by Flamingo Awadalla et al. (2023) as Encoder for the time series patches, yielding a fixed-size latent representation:

$$\mathbf{Z}_{\text{latent}} = \text{PerceiverResampler}(\mathbf{E}_{1:N}) \in \mathbb{R}^{N_{\text{latent}} \times d_{\text{time}}}, \tag{4}$$

Here, $d_{\text{time}}$ is the dimensionality of the time-series features by the perceiver, in our case $(N, 1)$, encoding one time series with one channel at a time.

**Text-Time-Series Gated Cross-Attention**   To integrate $\mathbf{Z}_{\text{latent}}$ into the LLM, we add gated cross-attention layers every $N$ (hyperparameter) transformer blocks which compute:

$$\mathbf{Q}_{\text{text}} = \mathbf{x}\mathbf{W}_Q, \quad \mathbf{K}_{\text{ts}} = \mathbf{Z}_{\text{latent}}\mathbf{W}_K, \quad \mathbf{V}_{\text{ts}} = \mathbf{Z}_{\text{latent}}\mathbf{W}_V \tag{5}$$

$$\text{GatedCrossAttention}(\mathbf{x}, \mathbf{Z}_{\text{latent}}) = x + \gamma \cdot \text{softmax}\left(\frac{\mathbf{Q}_{\text{text}}\mathbf{K}_{\text{ts}}^T}{\sqrt{d_k}}\right)\mathbf{V}_{\text{ts}}. \tag{6}$$

where $\gamma_{\text{attn}}$ is a learnable parameter controlling the influence of the time-series, $\mathbf{x} \in \mathbb{R}^{T \times d_{\text{model}}}$, the LLM input, $\mathbf{W}_Q, \mathbf{W}_K, \mathbf{W}_V \in \mathbb{R}^{d_{\text{model}} \times d_k}$ learned projection matrices, and $d_k$ the key dimension.

**Conditioning Text-tokens on Time-Series via Special Tokens**  The LLM processes tokens autoregressively, attending to previous inputs. Following OpenFlamingo Awadalla et al. (2023), we introduce special tokens $\langle \text{TS} \rangle$ and $\langle \text{endofchunk} \rangle$ to indicate when time series modalities should be incorporated. Upon encountering $\langle \text{TS} \rangle$, the model conditions on the corresponding latent representation $\mathbf{Z}_{\text{latent}}$ via gated cross-attention. A typical input prompt is

$$\mathbf{X}_{\text{input}} = [\text{pre\_prompt}, \langle \text{TS} \rangle, \text{ts\_desc}_1, \langle \text{endofchunk} \rangle, \langle \text{TS} \rangle, \text{ts\_desc}_2, \langle \text{endofchunk} \rangle, \text{post\_prompt}]$$
(7)

where $\langle \text{TS} \rangle$ triggers multimodal conditioning and $\langle \text{endofchunk} \rangle$ signals the end of text describing a time series. This setup enables interleaving multiple text and time series segments Awadalla et al. (2023). The embeddings of the special tokens are learned during training.

## 4 Experiments

In the following, we outline our training methodology and report results on multiple-choice Time Series Question Answering (TSQA) and time-series reasoning datasets. We compare OpenTSLM-SoftPrompt and OpenTSLM-Flamingo against each other and baselines in terms of performance, and report video random access memory (VRAM) requirements for training OpenTSLM. We present sample model outputs across datasets and an evaluation for ECG rationales by medical doctors.

### 4.1 Multi-Stage Curriculum Learning – Teaching LLMs Time Series

Following Chow et al. (2024), we adopt a two-stage curriculum to train TSLMs. In stage one (encoder warmup), we use two synthetic time-series datasets to pretrain the encoder:
- **TSQA Wang et al. (2024)** Multiple-choice time-series question answering on synthetic data for learning simple temporal patterns (e.g., ascending/descending trends).
- **Time-Series Captioning (M4-Captions)** We generate pseudo-labeled captions using ChatGPT, prompted with plots of time series of the M4 dataset Makridakis et al. (2020) (see Section A.5.1).

In stage two, we introduce three new CoT time-series datasets covering human activity recognition (HAR), sleep staging, and electrocardiogram (ECG) Question Answering (QA). We generated these using GPT-4o by providing a plot and ground-truth answer for each sample, then asking the model to produce rationales leading to the correct response. Further details are provided in Section A.2.
- **HAR-CoT** three-axis accelerometer data combined from DaLiAc Leutheuser et al. (2013), DOMINO Arrotta et al. (2023), HHAR Stisen et al. (2015), PAMAP2 Reiss & Stricker (2012), RealWorld Sztyler & Stuckenschmidt (2016), and datasets from Shoaib et al. (2013; 2014; 2016). Sampled at 50 Hz, split into 2.56s windows, 8 activities: sitting, standing, lying, walking, running, biking, walking upstairs, walking downstairs. See Section A.2.1 for detailed description.
- **Sleep-CoT** Based on SleepEDF Kemp et al. (2000); Goldberger et al. (2000), using 30s electroencephalogram (EEG) segments for sleep staging. Following prior work Chow et al. (2024); Pouliou et al. (2025), Non-rapid eye movement (REM) stages 3 and 4 are merged, yielding five classes: Wake, REM, Non-REM1, Non-REM2, Non-REM3. See Section A.2.2 for details.
- **ECG-QA-CoT** Based on ECG-QA Oh et al. (2023), which provides 12-lead 10s ECGs and clinical context, we excluded comparison questions, retaining 42/70 templates. This yielded 3,138 unique questions across 240k samples (see Section A.2.3).

All datasets are split into **80/10/10 train/validation/test** sets. Table 3 in Section A.1 summarizes number of samples in the datasets, number of time series and lengths.

**Training objective**  In all stages, we frame the task as an autoregressive language modeling problem. During training and evaluation, the model is prompted to generate outputs in a structured format, consisting of a free-form rationale followed by the final prediction: ``<reasoning> Answer: <final answer>''. Formally, the loss is defined by Equation 8, where $\mathbf{Z}_{\text{ts}}$ are the

$$\mathcal{L}_{\text{LM}} = -\sum_{t=1}^{T} \log P(y_t \mid y_{<t}, \mathbf{x}_{1:t}, \mathbf{Z}_{\text{ts}}; \Theta)$$
(8)

time-series features, and $\Theta$ the learnable weights, i.e., the TimeSeriesEncoder, MLP, and LoRA in OpenTSLM-SoftPrompt, and TimeSeriesEncoder and cross-attention in OpenTSLM-Flamingo.

### 4.2 Baselines

We compare OpenTSLM against three baselines using the same open-weight LLMs, i.e., Llama-3.2(1B, 3B) and Gemma3 (270M, 1B-PT), and additionally GPT-4o (gpt-4o-2024-08-06).

1. **Tokenized time-series**: Using the open-source code provided by Gruver et al. (2023), we tokenize time series into text inputs and report zero-shot performance on the test set.

2. **Tokenized finetuned**: Same as 1. (excluding GPT-4o), but finetuned with LoRA Hu et al. (2021) on the training set. We choose best model by validation loss, and report performance on test set.

3. **Image (Plot)**: We convert time series into plots and provide them as input to GPT-4o and Gemma-4b-pt (since the smaller Gemma 3 variants do not support image input).

4. **Random baseline**: For comparison, we report the expected performance of a predictor that selects labels uniformly at random, adjusted to each dataset's label distribution.

### 4.3 QUANTITATIVE RESULTS ON TIME-SERIES CLASSIFICATION

We present performance on the test splits of TSQA, HAR-CoT, Sleep-CoT, and ECG-QA-CoT and report macro-F1 score and accuracy in Table 2. OpenTSLM models achieve the highest performance

Table 2: Performance comparison on time series question answering (TSQA) and time series reasoning (HAR-CoT, Sleep-CoT, ECG-QA-CoT) tasks between OpenTSLM models and baselines.

| Method | Model | TSQA | | HAR-CoT | | Sleep-CoT | | ECG-QA-CoT | |
|---|---|---|---|---|---|---|---|---|---|
| | | F1 | Acc | F1 | Acc | F1 | Acc | F1 | Acc |
| Random Baseline | | 33.33 | 33.33 | 11.49 | 12.50 | 17.48 | 20.00 | 16.47 | 20.18 |
| Tokenized Time-Series | Llama3.2-1B | 16.01 | 31.04 | 0.00[*1] | 0.00 | 2.14 | 0.65 | 0.00 | 0.00 |
| | Llama3.2-3B | 16.24 | 32.06 | 0.00 | 0.00 | 5.66 | 12.15 | 0.00 | 0.00 |
| | Gemma3-270M | 10.52 | 9.58 | 0.00 | 0.00 | 0.00 | 0.00 | 0.00 | 0.00 |
| | Gemma3-1B-pt | 11.76 | 12.92 | 0.00 | 0.00 | 0.00 | 0.00 | 0.00 | 0.00 |
| | GPT-4o | 45.32 | 45.29 | 2.95 | 11.74 | 15.47 | 16.02 | 18.19 | 28.76 |
| Tokenized Finetuned | Llama3.2-1B | 83.74 | 81.40 | 51.28 | 62.71 | 9.05 | 24.19 | OOM[*2] | OOM |
| | Llama3.2-3B | 84.54 | 82.06 | 60.44 | 66.87 | 5.86 | 14.30 | OOM | OOM |
| | Gemma3-270M | 68.05 | 65.40 | 40.66 | 54.56 | 0.00 | 0.00 | OOM | OOM |
| | Gemma3-1B-pt | 82.85 | 83.42 | 52.15 | 63.90 | 0.00 | 0.00 | OOM | OOM |
| Image (Plot) | Gemma3-4B-pt | 48.77 | 50.60 | 1.72 | 0.89 | 6.75 | 14.95 | 1.90 | 1.03 |
| | GPT-4o | 59.24 | 62.10 | 10.83 | 13.90 | 4.82 | 10.75 | 24.95 | 33.30 |
| OpenTSLM SoftPrompt | Llama3.2-1B | **97.50** | **97.54** | **65.44** | **71.48** | **69.88** | **81.08** | 32.84 | 35.49 |
| | Llama3.2-3B | 97.37 | 97.33 | 64.87 | 67.89 | 54.40 | 72.04 | 33.67 | 36.25 |
| | Gemma3-270M | 40.32 | 26.79 | 1.43 | 0.55 | 7.96 | 5.91 | 1.29 | 1.11 |
| | Gemma3-1B-pt | 87.29 | 89.18 | 40.52 | 45.17 | 30.99 | 36.56 | 27.86 | 34.76 |
| OpenTSLM Flamingo | Llama3.2-1B | 94.08 | 94.00 | 62.93 | 69.27 | 49.33 | 67.31 | 34.62 | 38.14 |
| | Llama3.2-3B | 90.14 | 90.10 | 62.77 | 69.03 | 45.45 | 69.14 | **40.25** | **46.25** |
| | Gemma3-270M | 77.86 | 78.12 | 57.75 | 63.43 | 51.38 | 68.49 | 32.71 | 35.50 |
| | Gemma3-1B-pt | 92.56 | 92.46 | **65.44** | **71.48** | 43.69 | 60.67 | 35.31 | 37.79 |

Note: Gemma models have smaller context than Llama (32k vs. 128k); softprompt uses up context, performing worse. [*1]0.00 model failed to produce "Answer: {answer}" template, often repeating input prompt (see Section A.4). [*2]OOM - Out of memory: 12 ECG leads of 10s tokenize to 80k tokens, requiring >100GB VRAM.

across benchmarks, while most tokenized text-only baselines fail to produce valid outputs, not answering in the expected template but merely repeating inputs or starting to count (see Section A.4), resulting in 0.00 F1 on HAR for all models except for GPT-4o (2.95). GPT-4o yields only 2.95 F1 with text but improves substantially with plots (e.g., 10.83 on HAR, 59.24 on TSQA). Gemma3-4b similarly achieves better results TSQA and Sleep-CoT (48.77 and 6.75). Llama models achieve 2.14 and 5.65F1 on Sleep, respectively, while Gemma models again achieve 0.00, likely due to their smaller context window (32k vs. 128k). By contrast, OpenTSLM–SoftPrompt with Llama3.2-1B attains 97.50 F1 score (97.54 accuracy) on TSQA, with Llama3.2-3B at 97.37 (97.33); Flamingo variants are close (e.g., Llama3.2-1B 94.08 (94.00)), while the strongest tokenized-finetuned baseline reaches 84.54 (82.06) and GPT-4o with image inputs at 59.24 (62.10). On HAR-CoT, the strongest results are 65.44F1 (71.48 accuracy) for OpenTSLM–SoftPrompt (Llama3.2-1B) and 65.44 (71.48) for OpenTSLM–Flamingo (Gemma3-1B-pt); the best tokenized-finetuned baseline records 60.44 (66.87). On Sleep-CoT, OpenTSLM–SoftPrompt (Llama3.2-1B) achieves 69.88 (81.08), followed by OpenTSLM–SoftPrompt (Llama3.2-3B) at 54.40 (72.04) and Flamingo (Gemma3-270M) at 51.38 (68.49); tokenized-finetuned baselines remain lower (best 9.05 (24.19)). On ECG-QA-CoT, OpenTSLM–Flamingo (Llama3.2-3B) leads with 40.25 (46.25).

## 4.4 EVALUATION OF MEMORY USE DURING TRAINING

We evaluate peak VRAM usage during training for both OpenTSLM variants. Figure 4 summarizes peak VRAM on TSQA, HAR–CoT, SleepEDF–CoT, and ECG–QA–CoT. OpenTSLM-Flamingo

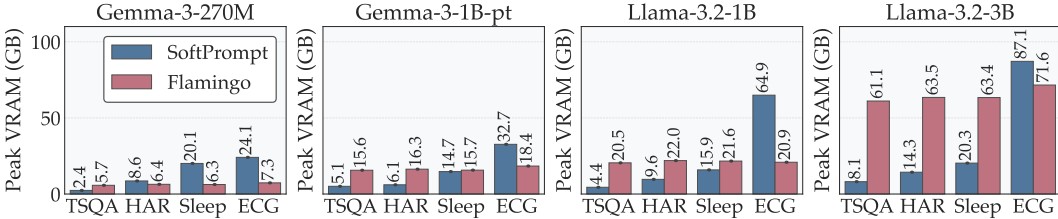

Figure 4: VRAM memory usage in training across datasets.

shows near-constant memory across datasets: Llama-3.2-1B requires around 20–22 GB and Llama-3.2-3B around 61–72 GB; Gemma-3-270M is 5.7–7.3 GB and Gemma-3-1B-pt 15.6–18.4 GB. In contrast, OpenTSLM-SoftPrompt vary substantially with the dataset: Llama-3.2-1B requires from 4.4 GB (TSQA) up to 64.9 GB (ECG–QA–CoT), and for Llama-3.2-3B from 8.1 GB to 87.1 GB; Gemma-3-270M spans 2.4–24.1 GB and Gemma-3-1B-pt 5.1–32.7 GB.

To further investigate memory scaling, we train models on a simulated dataset (see Section A.7.2) with random inputs of shape $(N \times L)$, where $N$ is the number of time series processed concurrently and $L$ the sequence length. We report max VRAM usage in Figure 5 (exact values are available in Table 10).

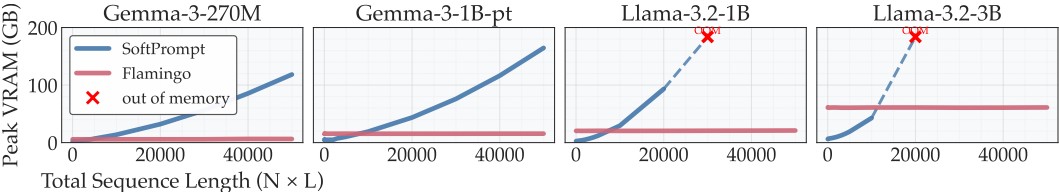

Figure 5: VRAM usage vs. total time-series size $N \times L$ (number of series × length)

VRAM for OpenTSLM-Flamingo effectively stays constant as $N$ increases from 1 to 5 and $L$ from 10 to 10,000 (e.g., Llama-1B $\approx$20.4–21.0 GB; Llama-3B $\approx$60.7–61.1 GB; Gemma-270M $\approx$5.7–6.4 GB; Gemma-1B $\approx$15.4–15.6 GB). By contrast, SoftPrompt scales with both $N$ and $L$ (see Figure 5 in Section A.7.2): for Llama-1B, VRAM rises from $\sim$2.6 GB at $L$=10, $N$=1 to $\sim$29.5 GB at $L$=10,000, $N$=1 and exceeds memory at $L$=10,000, $N\geq$3; Llama-3B shows a similar pattern (6.3 GB $\rightarrow$ 42.7 GB at $N$=1, OOM by $N\geq$3). Gemma-270M and Gemma-1B reach up to $\sim$118 GB and $\sim$165 GB, respectively, at $L$=10,000, $N$=5.

## 4.5 QUALITATIVE RESULTS AND EXPERT EVALUATION OF ECG RATIONALES

Both **OpenTSLM** variants remain text models, trained to generate rationales for classification rather than outputting only a class label. Figure 6a shows example rationales for *human activity recognition*, Figure 6b for ECG-QA, and Figure 6c for sleep staging. Figure 6d shows time series captioning on M4 data.

To evaluate the quality of model rationales, we conducted an expert review with five cardiologists from Stanford Hospital on rationales generated by OpenTSLM-Flamingo-Llama3.2-3B (best model) for ECG-QA. We randomly sampled two examples per template (84 total), each reviewed by at least two cardiologists. Evaluation followed a rubric derived from the American College of Cardiology/American Heart Association Clinical Competence Statement on ECGs Pangaro (1999); Committee Members et al. (2001) and based on the RIME ("Reporter–Interpreter–Manager–Educator") framework Pangaro (1999) (see A.6), assessing whether the model: (1) correctly identified relevant ECG features; (2) appropriately connected them to the final answer; (3) incorporated patient context (age, artifacts, ...). Overall, the model gave a correct or partially correct ECG interpretation in 92.9% of cases, spanning ECG recognition, reasoning, and contextualization. OpenTSLM showed strongest performance in clinical context integration (85.1% positive) compared to ECG pattern recognition (65.5% positive) and clinical reasoning (62.5% positive) (Figure 7a). Assessment

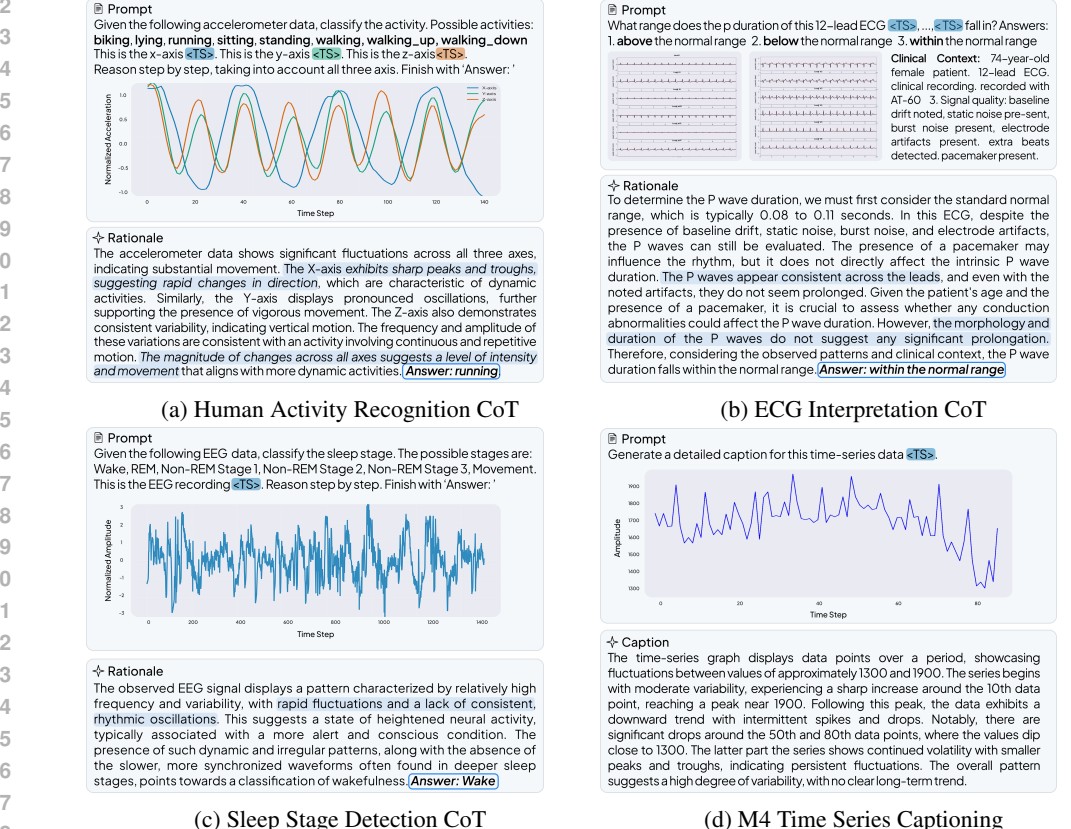

(a) Human Activity Recognition CoT

(b) ECG Interpretation CoT

(c) Sleep Stage Detection CoT

(d) M4 Time Series Captioning

Figure 6: Example CoT rationales for HAR, Sleep Staging, ECG-QA and M4 captioning, generated with OpenTSLM-Flamingo/Llama3.2-1B. More examples are provided in Section A.5.

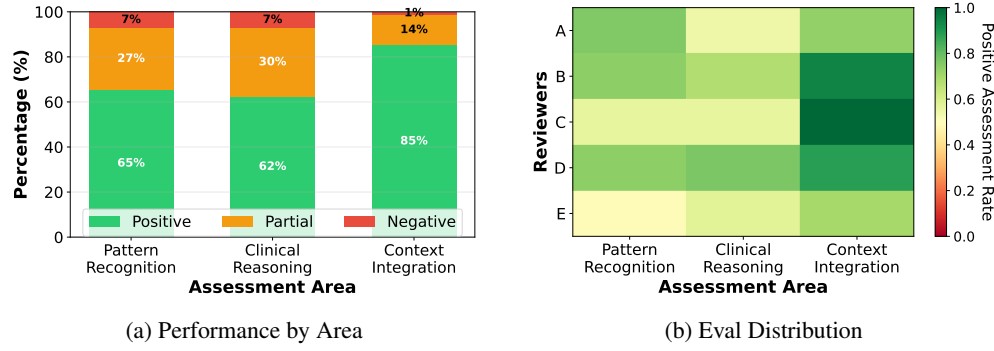

(a) Performance by Area

(b) Eval Distribution

Figure 7: Qualitative evaluation of CoT rationales and inter-reviewer agreement patterns.

patterns varied notably across reviewers, with some reviewers consistently more favorable across all evaluation areas (Figure 7b). Reviewer disagreement was most common for clinical reasoning, where moderate disagreements were observed between adjacent assessment categories. Complete disagreements between positive and negative assessments were relatively rare across all areas (Figure 17 in Appendix A.6).

## 5 DISCUSSION

All OpenTSLM models consistently outperform baselines. Text-only models often fail to follow the answer template and thus perform at or below chance (Section 4.1). Finetuned baselines improve substantially on HAR-CoT (60.44% F1 vs. 0% for Llama-3.2-1B) but only slightly on Sleep-CoT (9.05 vs. 2.14). ECG-QA finetuning was infeasible due to high VRAM demands (80k tokens require

>100GB per sample). OpenTSLM-SoftPrompt performs best on shorter sequences (Sleep-CoT, TSQA) but becomes impractical as VRAM requirements grow with sequence length (>180GB in simulations with 10,000-length series). With softprompting, smaller models like Gemma-3 270M and 1B quickly exhaust their context and underperform. In contrast, OpenTSLM-Flamingo sustains stable memory across sequence lengths and series (up to 60GB for Llama-3.2-3B with five 10,000-length series). This allows even tiny models, such as Gemma-270M, to deliver strong results, highlighting the efficiency of cross-attention for treating time series as a native modality.

**Practical implications.** Our results show that even frontier LLMs like GPT-4o are poorly suited for time-series reasoning and that time series must be treated as a distinct modality. With OpenTSLM, even small models like Gemma3 270M outperform GPT-4o (∼200B parameters Abacha et al. (2025)) at a fraction of the compute and cost, enabling efficient on-device or mobile deployment. We recommend using OpenTSLM-SoftPrompt for short time series, where it delivers strong performance while requiring only a small number of additional parameters during finetuning. However, because SoftPrompt's memory usage grows exponentially with sequence length, it becomes impractical for longer horizons or multi-series inputs. In contrast, we recommend OpenTSLM-Flamingo for longer time-series and multivariate sensor data (e.g, 12-lead ECG, 3-axis IMU) and as a general-purpose solution, as it maintains nearly constant memory consumption across extended or multi-series contexts, and offers better performance on complex datasets (like ECG-QA). Perhaps the greatest advantage of TSLMs is the interface they provide for contextualizing results. In ECG-QA, OpenTSLM correctly identified the relevant ECG features in most cases, with missing context only 7.1% of the time. The model demonstrated particularly strong clinical context integration (85.1% positive assessments), thereby offering clinicians and researchers a transparent window into the model's reasoning. As trust is important in medicine, this transparency underscores the value of applying LLMs to time series.

**Comparison with prior work.** Our approach differs from prior work in several ways. First, we introduce time series as a new modality for LLMs, unlike Sivarajkumar & Wang (2023) and Kim et al. (2024), which tokenize time series. Second, we frame tasks as joint text–time-series reasoning, training models to generate rationales that integrate temporal information. This contrasts with Med-ualTime Ye et al. (2025) and Time2Lang Pillai et al. (2025), which reprogrammed LLMs with fixed classification or forecasting heads, removing language generation capabilities. Notably, OpenTSLM achieves 40.25 F1 on ECG-QA-CoT, producing rationales across 3,138 questions and 42 templates with diverse answer options. By comparison, Ye et al. report 76 F1 on PTB-XL (underlying dataset of ECG-QA) with only four classes and a fixed classification head Ye et al. (2025). Third, unlike SensorLM Zhang et al. (2025), which is trained from scratch, our models build on pretrained open-weight LLMs, retaining pretrained knowledge. Fourth, while prior work used soft prompting Chow et al. (2024); Wang et al. (2025) to model time series implicitly by concatenating text-tokens with derived time-series tokens, we find that this approach scales poorly in memory use. In contrast, our OpenTSLM-Flamingo approach models time series explicitly via a separate encoding integrated via cross-attention, scaling better to long sequences.

**Limitations.** We acknowledge several limitations. First, our method of encoding time series may not be optimal, as we rely on including mean and standard deviation in accompanying texts to preserve temporal scale. Second, we generated CoT datasets using GPT-4o on plots, which we have shown to perform poorly on these plots alone. Curated datasets likely lead to better rationales. Third, framing tasks as natural language generation does not ensure that the model prioritizes the correct label, underscoring the need for loss functions that explicitly enforce correct answers. Fourth, we did not conduct ablation studies; for example, although OpenTSLM-Flamingo introduces gated cross-attention layers between every two transformer blocks, comparable performance might be achievable with fewer. Finally, while we report strong results on individual datasets, we have not yet demonstrated generalization to unseen data, an essential step toward general TSLMs.

## 6 CONCLUSION

Our results show that both OpenTSLM variants enable small-scale LLMs to outperform much larger text-only models on time-series tasks, demonstrating that lightweight, domain-adapted architectures can achieve strong performance without massive model scales. With OpenTSLM, we extend open-weight pretrained LLMs to process time series retaining knowledge while adapting them to temporal domains. This work may lay the foundation for general-purpose TSLMs capable of handling diverse time-series datasets. Although our focus is healthcare, the ability to reason over longitudinal data has broad relevance in domains such as finance, supply chain management, and industrial monitoring.

## REPRODUCIBILITY STATEMENT

All source code associated with this work is publicly available. All external datasets used are open source, and any datasets generated by us have also been released as open source. We additionally release all trained model weights. We also provide the notebooks annotated by clinical doctors for rationale generation on the ECG-QA dataset. These resources ensure full reproducibility of our results.

## USE OF LARGE LANGUAGE MODELS

Large Language Models (LLMs) were partially used for text editing, in limited instances, to improve the grammar and clarity of the original text. LLMs were additionally used for reviewing parts of the source code to identify critical errors or bugs. No LLMs were used for data analysis, experimental design, or drawing scientific conclusions.

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

# A  CHATAPPENDIX

## A.1  TRAINING DETAILS

Table Table 3 provides an overview of the datasets used during training. All data was split into ratios

| | Dataset | #Samples (Train/Val/Test) | Num series | Length | Frequency |
|---|---|---|---|---|---|
| **Stage 1** | TSQA[*1] | 38,400 / 4,800 / 4,800 | 1 | Hours to Years | Not specified |
| | M4-Captions | 80,000 / 10,000 / 10,000 | 1 | 64-512 points | Not specified |
| **Stage 2** | HAR-CoT | 68,542 / 8,718 / 8,222 | 3 | 2.56s | 50Hz |
| | Sleep-CoT | 7,434 / 930 / 930 | 1 | 30s | 100Hz |
| | ECG-QA-CoT | 159,313 / 31,137 / 41,093 | 12 | 10s | 100Hz |

Table 3: [*1]TSQA Wang et al. (2024) Overview of datasets used in Stage 1 (pretraining tasks) and Stage 2 (task-specific CoT reasoning). Datasets are split in 80/10/10 ration.

of 80/10/10 for train/val/test sets.

### A.1.1  TRAINING CONFIGURATION

The models were trained with the following configuration:

- **Optimizer:** AdamW
- **Learning Rates:**
    - **OpenTSLM-SP:**
        * Time series encoder: $2 \times 10^{-4}$
        * LoRA: $2 \times 10^{-4}$
        * Projector: $1 \times 10^{-4}$
    - **OpenTSLM-Flamingo:**
        * Encoder: $2 \times 10^{-4}$
        * Cross-attention layers: $2 \times 10^{-4}$
- **Scheduler:** Linear learning rate schedule with warmup
- **Warmup:** $10\%$ of total training steps
- **Gradient Clipping:** $\ell_2$-norm capped at 1.0
- **Weight Decay:** 0.01
- **Training Length:** Up to 200 epochs with early stopping (patience = 5 epochs)

Learning rate choices were informed by Chow et al. (2024).

## A.2  GENERATION OF MULTIVARIATE TIME SERIES COT DATASETS

This section provides detailed descriptions of the CoT datasets generated for our study: Human Activity Recognition (HAR-CoT), Sleep Stage Classification (SleepEDF-CoT), and Electrocardiogram Question Answering (ECG-QA-CoT).

Our objective was to enable TSLMs not only to classify time series but also to generate explicit reasoning chains. Since few datasets include CoT text, we generated our own multivariate time series CoT datasets using widely adopted benchmarks in HAR, sleep staging, and ECG-QA, following a similiar approach as proposed by Chow et al. (2024).

For each dataset, we generated rationales with GPT-4o by providing a plot of the data along with the correct label, and prompting the model to produce a rationale leading to that label. The exact prompts are described in Sections A.2.1, A.2.2, and A.2.3. We carefully engineered the prompts and manually reviewed a subset of samples to ensure the generated rationales were consistent and sensible. When plotting, original data was used without normalization. If multiple time series were present in a sample (e.g., three in HAR or twelve in ECG), all were plotted as separate subplots but combined into a single figure.

- **GPT-4o snapshot:** gpt-4o-2024-08-06
- **Temperature:** 0.3
- **Seed:** 42

The following subsections describe dataset-specific methodologies, data processing, prompts, answer selection, and final class distributions.

### A.2.1 HUMAN ACTIVITY RECOGNITON (HAR) CoT

We merged multiple HAR datasets spanning DaLiAc Leutheuser et al. (2013), DOMINO Arrotta et al. (2023), HHAR Stisen et al. (2015), PAMAP2 Reiss & Stricker (2012), RealWorld Sztyler & Stuckenschmidt (2016), and datstes from Shoaib et al. (2013; 2014; 2016). We retain only those activity classes present in all datasets. The final dataset includes eight activity classes: sitting, walking, standing, running, walking up stairs, walking down stairs, lying, and biking. The data is split into 2.56 second windows.

**Data Processing**  The dataset was processed to create 2.56-second windows of triaxial accelerometer data (X, Y, Z axes). Each sample was visualized as a multi-panel plot showing the acceleration signals across all three axes over the time window.

**Prompt for CoT generation**  We generated CoT rationales by prompting the model with a correct and dissimilar label. The following prompt template was used for HAR-CoT generation:

```
You are shown a time-series plot of accelerometer over a 2.56 second
window.
This data corresponds to one of two possible activities:
[CORRECT_ACTIVITY]
[DISSIMILAR_ACTIVITY]

Your task is to classify the activity based on analysis of the data.

Instructions:
- Begin by analyzing the time series without assuming a specific label.
- Think step-by-step about what the observed patterns suggest regarding
movement intensity and behavior.
- Write your rationale as a single, natural paragraph, do not use bullet
  points, numbered steps, or section headings.
- Do not refer back to the plot or to the act of visual analysis in your
rationale; the plot is only for reference but you should reason about the
  time-series data.
- Do **not** assume any answer at the beginning, analyze as if you do not
  yet know which class is correct.
- Do **not** mention either class label until the final sentence.
- Make sure that your last word is the answer. You MUST end your response
  with "Answer: [CORRECT_ACTIVITY]":
```

**Answer Selection Strategy**  For each sample, we implemented a dissimilarity-based answer selection strategy. Given a correct activity label, we selected the most dissimilar activity from a predefined mapping:

- **Sitting**: walking, running, biking, walking up, walking down
- **Walking**: sitting, lying, standing, biking, running
- **Standing**: walking, running, biking, walking up, walking down
- **Running**: sitting, lying, standing, biking, walking
- **Walking up**: sitting, lying, standing, biking, running
- **Walking down**: sitting, lying, standing, biking, running
- **Lying**: walking, running, biking, walking up, walking down
- **Biking**: sitting, lying, standing, walking, running

This strategy ensured that the binary classification tasks were challenging and required genuine analysis of movement patterns rather than simple pattern recognition.

**Label distribution**

### A.2.2 SLEEP STAGE CLASSIFICATION CHAIN-OF-THOUGHT (SLEEPEDF-CoT)

The SleepEDF-CoT dataset was generated from the Sleep-EDF database, which contains polysomnography recordings with expert-annotated sleep stage labels. The dataset includes five sleep stages: Wake (W), Non-REM stage 1 (N1), Non-REM stage 2 (N2), Non-REM stage 3 (N3), and REM sleep (REM).

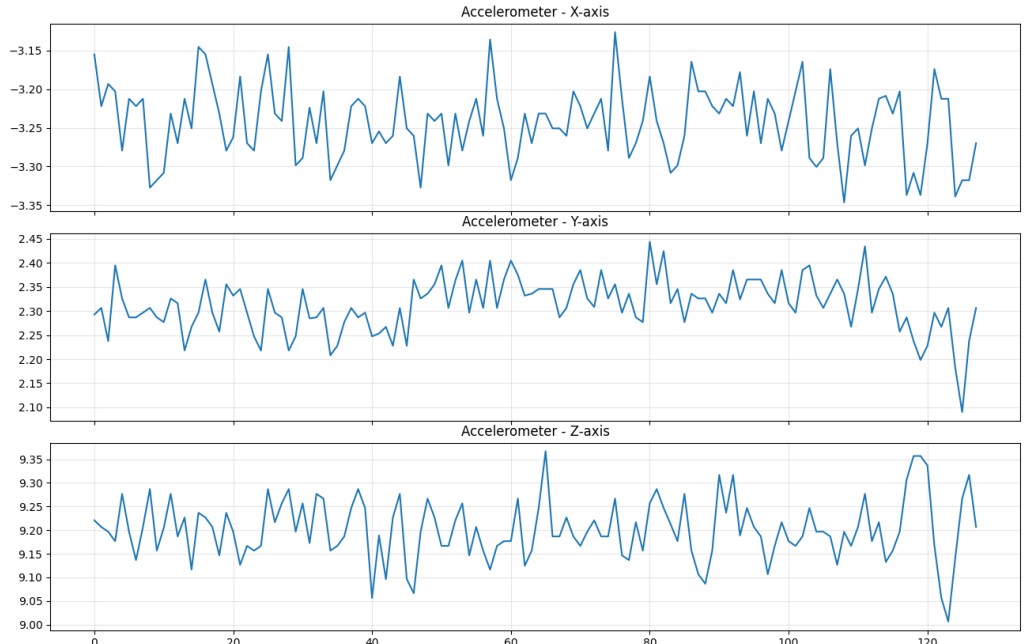

Figure 8: Sample HAR signal input to GPT-4o for rationale generation

Table 4: Per-class sample distribution for HAR-CoT train, validation, and test sets

| Class | Train (n=68542) | Val (n=8718) | Test (n=8222) |
|---|---|---|---|
| Biking | 4037 (5.9%) | 435 (5.0%) | 473 (5.8%) |
| Lying | 4305 (6.3%) | 682 (7.8%) | 444 (5.4%) |
| Running | 8101 (11.8%) | 948 (10.9%) | 1057 (12.9%) |
| Sitting | 18997 (27.7%) | 2315 (26.6%) | 2342 (28.5%) |
| Standing | 11001 (16.1%) | 1449 (16.6%) | 1264 (15.4%) |
| Walking | 12675 (18.5%) | 1611 (18.5%) | 1508 (18.3%) |
| Walking Down | 4514 (6.6%) | 710 (8.1%) | 542 (6.6%) |
| Walking Up | 4912 (7.2%) | 568 (6.5%) | 592 (7.2%) |

**Data Processing** The dataset was processed to create 30-second windows of EEG data from the Fpz-Cz channel. Each sample was visualized as a single-channel EEG plot showing brain activity patterns characteristic of different sleep stages.

**Prompt for CoT generation** We generated CoT rationales by prompting the model with a correct and dissimilar label. The following prompt template was used for SleepEDF-CoT generation:

```
You are presented with a time-series plot showing EEG data collected over
 a 30-second interval. This signal corresponds to one of two possible
sleep stages:
- [SLEEP_STAGE_1]
- [SLEEP_STAGE_2]

Your task is to determine the correct sleep stage based solely on the
observed patterns in the time series.

Instructions:
- Analyze the data objectively without presuming a particular label.
- Reason carefully and methodically about what the signal patterns
suggest
  regarding sleep stage.
```

```
- Write your reasoning as a single, coherent paragraph. Do not use bullet
  points, lists, or section headers.
- Do not reference the plot, visuals, or the process of viewing the data
  in your explanation; focus only on the characteristics of the time series
  .
- Do not mention or speculate about either class during the rationale,
  only reveal the correct class at the very end.
- Never state that you are uncertain or unable to classify the data. You
  must always provide a rationale and a final answer.
- Your final sentence must conclude with: "Answer: [CORRECT_SLEEP_STAGE]"
```

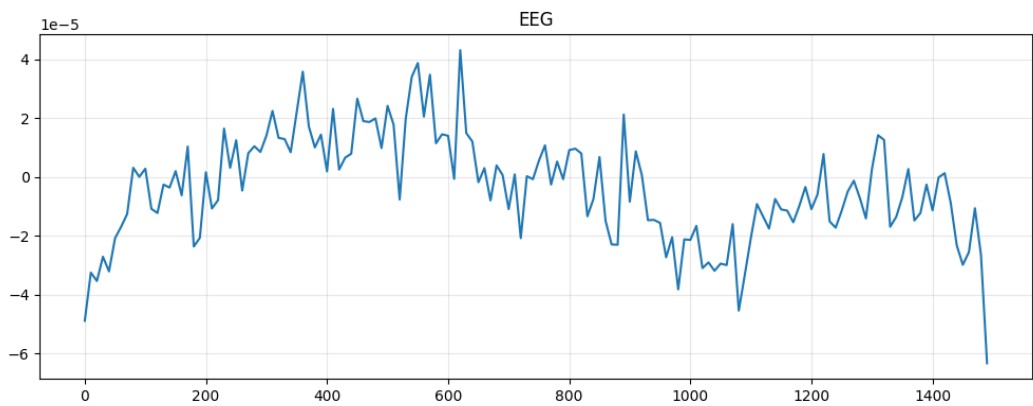

Figure 9: Sample EEG signal input to GPT-4o for sleep stage rationale generation

**Answer Selection Strategy**    For sleep stage classification, we implemented a dissimilarity-based strategy that pairs physiologically distinct sleep stages:

- **Wake (W)**: N3, N4, REM
- **N1**: W, N3, N4
- **N2**: W, REM
- **N3**: W, REM
- **N4**: W, REM
- **REM**: N2, N3, N4

This approach ensured that the binary classification tasks required understanding of fundamental differences in brain activity patterns between sleep stages.

**Label distribution**    SleepEDF dataset

Table 5: Per-class sample distribution for train, validation, and test sets (Sleep stages)

| Label | Train (n=7434) | Val (n=930) | Test (n=930) |
|---|---|---|---|
| Non-REM 1 | 410 (5.5%) | 52 (5.6%) | 51 (5.5%) |
| Non-REM 2 | 2057 (27.7%) | 257 (27.6%) | 257 (27.6%) |
| Non-REM 3 | 357 (4.8%) | 45 (4.8%) | 45 (4.8%) |
| Non-REM 4 | 299 (4.0%) | 37 (4.0%) | 38 (4.1%) |
| REM | 944 (12.7%) | 118 (12.7%) | 118 (12.7%) |
| Wake | 3367 (45.3%) | 421 (45.3%) | 421 (45.3%) |

### A.2.3 ELECTROCARDIOGRAM QUESTION ANSWERING CHAIN-OF-THOUGHT (ECG-QA-CoT)

The ECG-QA-CoT dataset was generated from the PTB-XL Wagner et al. (2020) database combined with the ECG-QA Oh et al. (2023) question templates. This dataset contains 12-lead ECG recordings with clinical questions covering various aspects of cardiac analysis, including rhythm analysis, morphology assessment, and diagnostic classification.

**Data Processing**   The dataset was processed to create complete 12-lead ECG recordings (I, II, III, aVR, aVL, aVF, V1, V2, V3, V4, V5, V6) sampled at 100 Hz. Each ECG was visualized as a multi-panel plot showing all 12 leads simultaneously, enabling comprehensive cardiac analysis.

**Prompt for CoT generation**   The following prompt template was used for ECG-QA-CoT generation:

```
You are presented with a complete 12-lead ECG recording showing all
standard leads (I, II, III, aVR, aVL, aVF, V1, V2, V3, V4, V5, V6).

Clinical Context: [CLINICAL_CONTEXT]

Question: [QUESTION]

This question has one of two possible answers:
- [ANSWER_OPTION_1]
- [ANSWER_OPTION_2]

Your task is to analyze the ECG and determine the correct answer based on
 the observed cardiac patterns. You may include the clinical context in
your analysis if it helps you determine the correct answer.

Instructions:
- Analyze the ECG systematically without presuming a particular answer.
- Consider rhythm, rate, morphology, intervals, and any abnormalities you
 observe across all 12 leads.
- Think step-by-step about what the ECG patterns indicate regarding the
clinical question above.
- Write your reasoning as a single, coherent paragraph. Do not use bullet
 points, lists, or section headers.
- Do not reference the visual aspects of viewing the ECG plot; focus on
the cardiac characteristics and clinical significance.
- Do not mention or assume either answer option during your rationale,
only reveal the correct answer at the very end.
- NEVER state uncertainty or inability to determine the answer. You MUST
always provide clinical reasoning and a definitive answer.
- Your final sentence must conclude with: "Answer: [CORRECT_ANSWER]"
```

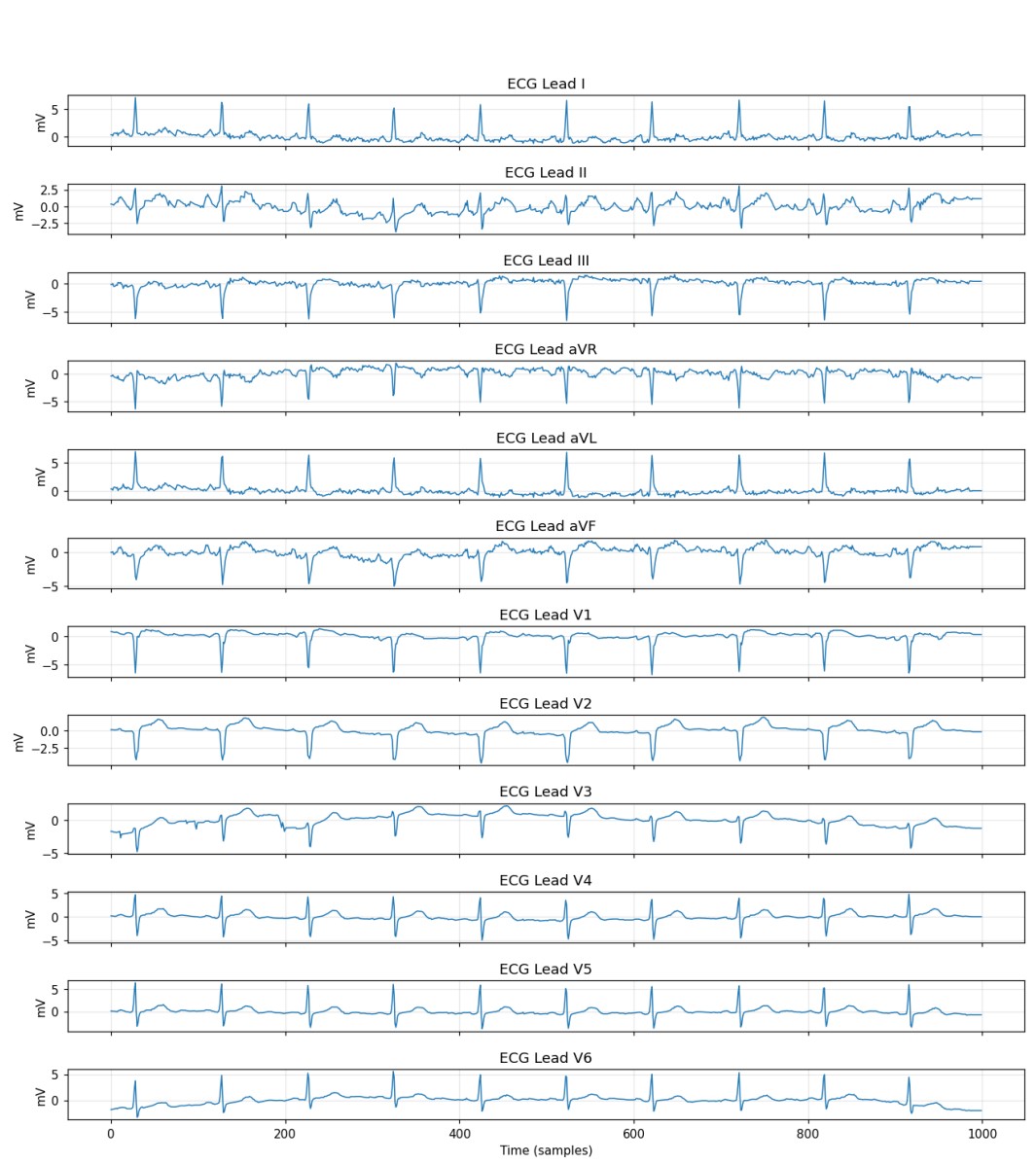

Figure 10: Sample ECG signal input to GPT-4o for rationale generation

Table 6: Per-template sample distribution for ECG-QA CoT train, validation, and test sets

| Template ID | Train (n=159,306) | Val (n=31,137) | Test (n=41,093) |
|---|---|---|---|
| Template 1 | 17,089 (10.7%) | 2,924 (9.4%) | 3,467 (8.4%) |
| Template 2 | 300 (0.2%) | 60 (0.2%) | 60 (0.1%) |
| Template 3 | 240 (0.2%) | 48 (0.2%) | 48 (0.1%) |
| Template 4 | 20,861 (13.1%) | 3,782 (12.1%) | 4,096 (10.0%) |
| Template 5 | 20,104 (12.6%) | 3,599 (11.6%) | 3,905 (9.5%) |
| Template 6 | 5,356 (3.4%) | 1,022 (3.3%) | 1,085 (2.6%) |
| Template 7 | 1,137 (0.7%) | 221 (0.7%) | 224 (0.5%) |
| Template 8 | 4,371 (2.7%) | 747 (2.4%) | 1,466 (3.6%) |
| Template 9 | 3,563 (2.2%) | 610 (2.0%) | 1,200 (2.9%) |
| Template 10 | 894 (0.6%) | 311 (1.0%) | 377 (0.9%) |
| Template 11 | 2,861 (1.8%) | 533 (1.7%) | 964 (2.3%) |
| Template 12 | 300 (0.2%) | 60 (0.2%) | 60 (0.1%) |
| Template 13 | 300 (0.2%) | 60 (0.2%) | 60 (0.1%) |
| Template 14 | 300 (0.2%) | 60 (0.2%) | 60 (0.1%) |
| Template 15 | 300 (0.2%) | 60 (0.2%) | 60 (0.1%) |
| Template 16 | 300 (0.2%) | 60 (0.2%) | 60 (0.1%) |
| Template 17 | 19,952 (12.5%) | 3,013 (9.7%) | 4,416 (10.7%) |
| Template 18 | 9,580 (6.0%) | 2,178 (7.0%) | 3,806 (9.3%) |
| Template 19 | 4,122 (2.6%) | 698 (2.2%) | 1,395 (3.4%) |
| Template 20 | 1,200 (0.8%) | 228 (0.7%) | 237 (0.6%) |
| Template 21 | 180 (0.1%) | 36 (0.1%) | 36 (0.1%) |
| Template 22 | 400 (0.3%) | 131 (0.4%) | 167 (0.4%) |
| Template 23 | 744 (0.5%) | 126 (0.4%) | 168 (0.4%) |
| Template 24 | 90 (0.1%) | 18 (0.1%) | 18 (0.0%) |
| Template 25 | 399 (0.3%) | 160 (0.5%) | 178 (0.4%) |
| Template 26 | 10,585 (6.6%) | 1,894 (6.1%) | 2,193 (5.3%) |
| Template 27 | 1,038 (0.7%) | 180 (0.6%) | 210 (0.5%) |
| Template 28 | 3,600 (2.3%) | 720 (2.3%) | 720 (1.8%) |
| Template 29 | 300 (0.2%) | 60 (0.2%) | 60 (0.1%) |
| Template 30 | 224 (0.1%) | 36 (0.1%) | 43 (0.1%) |
| Template 31 | 1,235 (0.8%) | 198 (0.6%) | 274 (0.7%) |
| Template 32 | 697 (0.4%) | 246 (0.8%) | 313 (0.8%) |
| Template 33 | 6,102 (3.8%) | 2,189 (7.0%) | 2,775 (6.8%) |
| Template 34 | 2,411 (1.5%) | 494 (1.6%) | 872 (2.1%) |
| Template 35 | 246 (0.2%) | 18 (0.1%) | 50 (0.1%) |
| Template 36 | 900 (0.6%) | 176 (0.6%) | 180 (0.4%) |
| Template 37 | 108 (0.1%) | 21 (0.1%) | 22 (0.1%) |
| Template 38 | 523 (0.3%) | 192 (0.6%) | 241 (0.6%) |
| Template 39 | 5,100 (3.2%) | 1,019 (3.3%) | 1,020 (2.5%) |
| Template 40 | 480 (0.3%) | 104 (0.3%) | 104 (0.3%) |
| Template 41 | 1,700 (1.1%) | 819 (2.6%) | 849 (2.1%) |
| Template 42 | 9,114 (5.7%) | 2,026 (6.5%) | 3,554 (8.6%) |

**Label distribution**

**Per-Template Label Distribution Summary**

| Template ID | Train Labels | Val Labels | Test Labels |
|---|---|---|---|
| Template 1 | no: 11360, yes: 4751, not sure: 978 | no: 1995, yes: 796, not sure: 133 | no: 2215, yes: 991, not sure: 261 |
| Template 2 | no: 200, yes: 100 | no: 40, yes: 20 | no: 40, yes: 20 |

| | | | |
|---|---|---|---|
| Template 3 | st/t change: 60, myocardial infarction: 60, none: 60, hypertrophy: 60, conduction disturbance: 60 | st/t change: 12, myocardial infarction: 12, none: 12, hypertrophy: 12, conduction disturbance: 12 | st/t change: 12, myocardial infarction: 12, none: 12, hypertrophy: 12, conduction disturbance: 12 |
| Template 4 | none: 6300, myocardial infarction in anteroseptal leads: 618, left anterior fascicular block: 593, myocardial infarction in inferior leads: 586, first degree av block: 585 | none: 1258, left ventricular hypertrophy: 110, myocardial infarction in anteroseptal leads: 109, left anterior fascicular block: 107, first degree av block: 107 | none: 1260, myocardial infarction in anteroseptal leads: 122, myocardial infarction in inferior leads: 118, left ventricular hypertrophy: 117, left anterior fascicular block: 117 |
| Template 5 | none: 6300, myocardial infarction in anteroseptal leads: 578, left anterior fascicular block: 565, first degree av block: 558, non-specific intraventricular conduction disturbance (block): 522 | none: 1248, left anterior fascicular block: 105, first degree av block: 103, myocardial infarction in anteroseptal leads: 99, left ventricular hypertrophy: 95 | none: 1260, myocardial infarction in anteroseptal leads: 117, left anterior fascicular block: 116, non-specific intraventricular conduction disturbance (block): 112, first degree av block: 109 |
| Template 6 | none: 1530, non-diagnostic t abnormalities: 306, ventricular premature complex: 300, non-specific st changes: 295, non-specific st depression: 294 | none: 306, non-specific st depression: 57, non-diagnostic t abnormalities: 56, ventricular premature complex: 55, voltage criteria (qrs) for left ventricular hypertrophy: 52 | none: 306, ventricular premature complex: 64, non-specific st depression: 63, non-diagnostic t abnormalities: 60, atrial premature complex: 60 |
| Template 7 | none: 360, bigeminal pattern (unknown origin, supraventricular, or ventricular): 105, atrial flutter: 99, sinus rhythm: 98, atrial fibrillation: 98 | none: 72, sinus rhythm: 19, bigeminal pattern (unknown origin, supraventricular, or ventricular): 19, atrial flutter: 18, atrial fibrillation: 17 | none: 72, bigeminal pattern (unknown origin, supraventricular, or ventricular): 21, sinus rhythm: 19, atrial fibrillation: 18, sinus tachycardia: 18 |
| Template 8 | myocardial infarction in anteroseptal leads: 1050, myocardial infarction in inferior leads: 830, left ventricular hypertrophy: 791, left anterior fascicular block: 705, non-specific ischemic: 512 | myocardial infarction in inferior leads: 130, left ventricular hypertrophy: 129, myocardial infarction in anteroseptal leads: 127, left anterior fascicular block: 114, none: 100 | myocardial infarction in anteroseptal leads: 304, left ventricular hypertrophy: 282, myocardial infarction in inferior leads: 259, left anterior fascicular block: 236, non-specific ischemic: 177 |
| Template 9 | myocardial infarction in anteroseptal leads: 635, left anterior fascicular block: 592, non-specific ischemic: 459, left ventricular hypertrophy: 432, first degree av block: 399 | left anterior fascicular block: 111, none: 100, non-diagnostic t abnormalities: 79, myocardial infarction in anteroseptal leads: 74, incomplete right bundle branch block: 70 | left anterior fascicular block: 206, myocardial infarction in anteroseptal leads: 194, non-specific ischemic: 155, left ventricular hypertrophy: 149, non-specific intraventricular conduction disturbance (block): 127 |

| | | | |
|---|---|---|---|
| Template 10 | none: 200, sinus rhythm: 135, atrial fibrillation: 118, sinus tachycardia: 108, sinus bradycardia: 107 | sinus rhythm: 56, none: 56, atrial fibrillation: 51, sinus tachycardia: 51, sinus arrhythmia: 42 | none: 100, sinus rhythm: 56, sinus tachycardia: 52, atrial fibrillation: 52, sinus bradycardia: 51 |
| Template 11 | non-specific st depression: 692, non-diagnostic t abnormalities: 570, ventricular premature complex: 414, low amplitude t-wave: 334, voltage criteria (qrs) for left ventricular hypertrophy: 329 | none: 100, non-diagnostic t abnormalities: 99, non-specific st depression: 81, ventricular premature complex: 64, abnormal qrs: 64 | non-specific st depression: 194, non-diagnostic t abnormalities: 182, ventricular premature complex: 142, voltage criteria (qrs) for left ventricular hypertrophy: 123, q waves present: 105 |
| Template 12 | no: 200, yes: 100 | no: 40, yes: 20 | no: 40, yes: 20 |
| Template 13 | no: 200, yes: 100 | no: 40, yes: 20 | no: 40, yes: 20 |
| Template 14 | no: 200, yes: 100 | no: 40, yes: 20 | no: 40, yes: 20 |
| Template 15 | no: 200, yes: 100 | no: 40, yes: 20 | no: 40, yes: 20 |
| Template 16 | no: 200, yes: 100 | no: 40, yes: 20 | no: 40, yes: 20 |
| Template 17 | no: 14455, yes: 5497 | no: 2270, yes: 743 | no: 3150, yes: 1266 |
| Template 18 | none: 2400, non-specific st depression: 1848, voltage criteria (qrs) for left ventricular hypertrophy: 1510, non-diagnostic t abnormalities: 1385, low amplitude t-wave: 1138 | none: 1150, non-specific st depression: 378, voltage criteria (qrs) for left ventricular hypertrophy: 216, q waves present: 114, non-diagnostic t abnormalities: 107 | none: 1200, voltage criteria (qrs) for left ventricular hypertrophy: 675, non-specific st depression: 645, non-diagnostic t abnormalities: 473, non-specific t-wave changes: 308 |
| Template 19 | none: 1695, lead I: 1509, lead V6: 1453, lead V5: 1322, lead aVL: 1242 | none: 415, lead I: 165, lead V6: 154, lead V5: 153, lead aVL: 138 | none: 655, lead I: 438, lead V6: 431, lead V5: 399, lead aVL: 392 |
| Template 20 | no: 800, yes: 400 | no: 160, yes: 68 | no: 160, yes: 77 |
| Template 21 | none: 60, left axis deviation: 30, right axis deviation: 30, extreme axis deviation: 30, normal heart axis: 30 | none: 12, left axis deviation: 6, right axis deviation: 6, extreme axis deviation: 6, normal heart axis: 6 | none: 12, left axis deviation: 6, right axis deviation: 6, extreme axis deviation: 6, normal heart axis: 6 |
| Template 22 | left axis deviation: 100, right axis deviation: 100, extreme axis deviation: 100, normal heart axis: 100 | left axis deviation: 50, normal heart axis: 50, right axis deviation: 23, extreme axis deviation: 8 | left axis deviation: 50, right axis deviation: 50, normal heart axis: 50, extreme axis deviation: 17 |
| Template 23 | no: 545, yes: 199 | no: 95, yes: 31 | no: 120, yes: 48 |
| Template 24 | none: 30, early stage of myocardial infarction: 20, middle stage of myocardial infarction: 20, old stage of myocardial infarction: 20 | none: 6, early stage of myocardial infarction: 4, middle stage of myocardial infarction: 4, old stage of myocardial infarction: 4 | none: 6, early stage of myocardial infarction: 4, middle stage of myocardial infarction: 4, old stage of myocardial infarction: 4 |
| Template 25 | none of myocardial infarction: 100, unknown stage of myocardial infarction: 100, middle stage of myocardial infarction: 100, early stage of myocardial infarction: 70, old stage of myocardial infarction: 29 | none of myocardial infarction: 50, unknown stage of myocardial infarction: 50, middle stage of myocardial infarction: 49, early stage of myocardial infarction: 6, old stage of myocardial infarction: 5 | none of myocardial infarction: 50, unknown stage of myocardial infarction: 50, middle stage of myocardial infarction: 50, early stage of myocardial infarction: 19, old stage of myocardial infarction: 9 |

| Template 26 | no: 7335, yes: 3250 | no: 1335, yes: 559 | no: 1470, yes: 723 |
|---|---|---|---|
| Template 27 | no: 715, yes: 323 | no: 120, yes: 60 | no: 145, yes: 65 |
| Template 28 | no: 2400, yes: 1200 | no: 480, yes: 240 | no: 480, yes: 240 |
| Template 29 | no: 200, yes: 100 | no: 40, yes: 20 | no: 40, yes: 20 |
| Template 30 | none: 60, baseline drift: 58, static noise: 56, burst noise: 50, electrodes problems: 44 | none: 12, baseline drift: 10, static noise: 10, burst noise: 10 | none: 12, static noise: 11, baseline drift: 10, burst noise: 10, electrodes problems: 7 |
| Template 31 | static noise: 448, none: 430, baseline drift: 333, burst noise: 309, electrodes problems: 17 | static noise: 95, none: 72, burst noise: 47, baseline drift: 45 | static noise: 99, none: 88, burst noise: 80, baseline drift: 71, electrodes problems: 1 |
| Template 32 | baseline drift: 252, static noise: 241, none: 200, burst noise: 174, electrodes problems: 23 | none: 100, static noise: 83, baseline drift: 78, burst noise: 22 | baseline drift: 112, static noise: 109, none: 100, burst noise: 58, electrodes problems: 5 |
| Template 33 | none: 2400, static noise: 1824, baseline drift: 1729, burst noise: 823, electrodes problems: 27 | none: 1200, static noise: 675, baseline drift: 358, burst noise: 79 | none: 1200, static noise: 744, baseline drift: 712, burst noise: 283, electrodes problems: 6 |
| Template 34 | lead III: 972, lead II: 904, lead I: 864, lead aVR: 844, lead aVL: 779 | none: 215, lead III: 182, lead II: 175, lead I: 169, lead aVR: 165 | lead III: 339, lead II: 327, lead I: 320, lead aVR: 305, lead aVL: 270 |
| Template 35 | no: 200, yes: 46 | no: 15, yes: 3 | no: 40, yes: 10 |
| Template 36 | no: 600, yes: 300 | no: 120, yes: 56 | no: 120, yes: 60 |
| Template 37 | supraventricular extrasystoles: 38, ventricular extrasystoles: 30, none: 30, extrasystoles: 28 | supraventricular extrasystoles: 7, extrasystoles: 6, none: 6, ventricular extrasystoles: 5 | supraventricular extrasystoles: 8, extrasystoles: 6, ventricular extrasystoles: 6, none: 6 |
| Template 38 | none: 200, supraventricular extrasystoles: 125, ventricular extrasystoles: 115, extrasystoles: 108 | none: 100, extrasystoles: 55, supraventricular extrasystoles: 27, ventricular extrasystoles: 16 | none: 100, supraventricular extrasystoles: 57, extrasystoles: 54, ventricular extrasystoles: 38 |
| Template 39 | no: 3400, yes: 1700 | no: 680, yes: 339 | no: 680, yes: 340 |
| Template 40 | none: 160, within the normal range: 110, above the normal range: 110, below the normal range: 100 | none: 36, within the normal range: 24, above the normal range: 24, below the normal range: 20 | none: 36, within the normal range: 24, above the normal range: 24, below the normal range: 20 |
| Template 41 | within the normal range: 600, above the normal range: 600, below the normal range: 500 | within the normal range: 300, above the normal range: 300, below the normal range: 219 | within the normal range: 300, above the normal range: 300, below the normal range: 249 |
| Template 42 | qt interval: 4393, rr interval: 4336, qt corrected: 4262, p duration: 4093, qrs duration: 4010 | rr interval: 902, qt interval: 880, qt corrected: 879, p duration: 872, qrs duration: 779 | rr interval: 1730, qt interval: 1672, p duration: 1614, qt corrected: 1592, qrs duration: 1486 |

### A.3   M4 Caption Dataset Generation

We constructed the M4-Caption dataset by pairing time series from the M4 forecasting competition dataset Makridakis et al. (2020) with model-generated natural language captions.

**Data processing**   We removed trailing padding from each tensor by truncating after the last non-zero element.

**Prompt for caption generation**   We combine a high-resolution plot, whose aspect ratio scales with sequence length to preserve visual fidelity and contextual detail, with the task to generate a detailed caption.

```
Generate a detailed caption for the following time-series data:
```

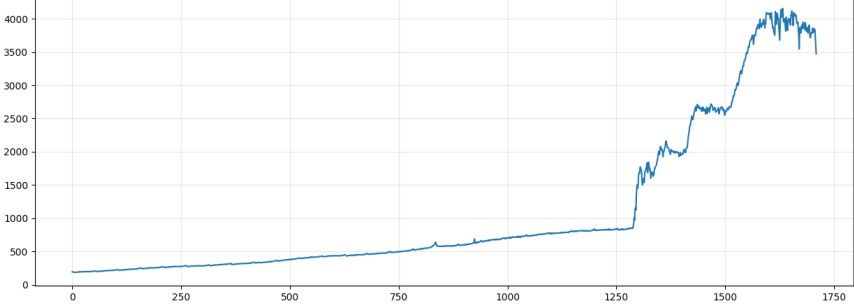

Figure 11: Sample M4 signal input to GPT-4o for caption generation

## A.4 EXAMPLE OF BASELINES FAILING TO PRODUCE MEANINGFUL OUTPUT

As shown in Table 2 in Appendix 4.3, some text-only models achieve 0% F1 score on the CoT datasets. This is because they fail to answer in the "$\langle rationale \rangle$ $Answer$ : $\langle answer \rangle$" template (see Appendix 4.1). We present some examples of such outputs in the following.

### A.4.1 LLAMA3.2-3B BASELINE OUTPUT ON HAR-CoT

INPUT PROMPT (TRUNCATED)

```
You are given accelerometer data in all three dimensions. Your task is
to classify the activity based on analysis of the data.

Instructions:
- Begin by analyzing the time series without assuming a specific label.
- Think step-by-step about what the observed patterns suggest regarding
movement intensity and behavior.
- Write your rationale as a single, natural paragraph, do not use bullet
points, numbered steps, or section headings.
- Do **not** mention any class label until the final sentence.

The following is the accelerometer data on the x-axis, it has mean
-3.2434 and std 0.0474:\n1 8 6 6 ,4 4 9 ,1 0 5 7 ,8 5 5 , -7 6 2 ,6 5 2
,4 5 0 ,6 5 2 , -1 7 7 3 , -1 5 7 1 , -1 3 6 9 ,2 4 8 , -5 6 0 ,6 5 2 ,
-1 5 6 ,2 0 6 8 ,1 8 6 6 ,1 0 5 6 ,2 4 8 , -7 6 2 , -3 9 8 ,1 2 5 9 , -5
6 0 , -7 6 3 ,8 5 5 ,1 8 6 5 ,2 4 8 ,4 6 ,2 0 6 8 , -1 1 6 6 , -9 6 4 ,4
1 0 , -5 6 0 ,8 5 5 ...
The following is the accelerometer data on the y-axis, it has mean 2.3132
 and std 0.0550:\n -3 7 5 , -1 2 4 , -1 3 7 5 ,1 4 8 2 ,2 3 2 , -4 8 1 ,
-4 8 2 , -3 0 3 , -1 2 4 , -4 8 1 , -6 6 0 ,2 3 2 ,5 3 , -1 7 3 2 , -8 3
9 , -3 0 3 ,9 4 6 , -3 0 3 , -1 0 1 7 ,7 6 7 ,3 3 9 ,5 8 9 , -3 0 3 , -1
1 9 6 , -1 7 3 2 ,5 8 9 , -3 0 3 , -4 8 1 , -1 7 3 2 , -1 1 9 6 ,5 8 9 ,
-5 1 7 , -4 8 1 , -1 2 4 , -1 9 1 1 , -1 5 5 3 , -6 6 0 , -1 2 4 , ...
The following is the accelerometer data on the z-axis, it has mean 9.2017
 and std 0.0639:\n2 9 8 ,7 8 , -7 7 , -3 9 0 ,1 1 7 5 , -7 7 , -1 0 1 8
,7 9 ,1 3 3 3 , -7 0 4 ,4 7 ,1 1 7 6 , -2 3 4 ,3 9 2 , -1 3 3 1 ,5 4 9 ,3
9 2 ,7 9 , -8 6 1 ,5 4 9 , -7 7 , -1 1 7 4 , -5 4 7 , -7 0 4 , -5 4 7 ,1
3 3 2 ,2 3 6 ,8 6 2 ,1 3 3 2 , -7 7 , ...

Possible activity labels are:
biking, lying, running, sitting, standing, walking, walking_down,
walking_up.

- Please now write your rationale. Make sure that your last word is the
answer. You MUST end your response with "Answer:"
```

GENERATED ANSWER

The accelerometer data on the x-axis has a mean of -3.2434 and a standard
 deviation of 0.0474. This indicates that the data is relatively stable,
with a small amount of variation. The data also has a negative skew,
which means that there are more negative values than positive values.
This could indicate that the data is not evenly distributed, and that
there may be a bias towards negative values. The accelerometer data on
the y-axis has a mean of 2.3132 and a standard deviation of 0.0550. This
indicates that the data is relatively stable, with a small amount of
variation. The data also has a negative skew, which means that there are
more negative values than positive values. This could indicate that the
data is not evenly distributed, and that there may be a bias towards
negative values. Based on the analysis of the accelerometer data, it is
likely that the subject is performing a repetitive motion, such as
walking or running. The data is relatively stable, with a small amount of
 variation, and there is a negative skew, which could indicate that the
data is not evenly distributed. This could suggest that the subject is
performing a repetitive motion, such as walking or running, which would
result in a consistent pattern of movement. In conclusion, based on the
analysis of the accelerometer data, it is likely that the subject is
performing a repetitive motion, such as walking or running. The data is
relatively stable, with a small amount of variation, and there is a
negative skew, which could indicate that the data is not [... repeated
multiple times]

### A.4.2 GEMMA3-1B-PT BASELINE OUTPUT ON SLEEP-COT

INPUT PROMPT (TRUNCATED)

You are given a 30-second EEG time series segment. Your task is to
classify the sleep stage based on analysis of the data.

Instructions:
- Analyze the data objectively without presuming a particular label.
- Reason carefully and methodically about what the signal patterns
suggest
 regarding sleep stage.
- Write your reasoning as a single, coherent paragraph.
- Only reveal the correct class at the very end.
- Never state that you are uncertain or unable to classify the data.
 You must always provide a rationale and a final answer.

The following is the EEG time series, it has mean 0.0000 and std 0.0000:\
n -3 2 0 , -1 8 0 ,5 5 , -2 1 1 , -6 5 , -4 6 6 , -2 6 9 , -4 4 1 , -7 5
9 , -8 2 9 , -1 0 6 5 , -8 2 9 , -1 0 7 7 , -8 2 9 , -8 9 3 , -7 2 7 , -1
 0 2 6 , -7 7 2 , -9 0 5 , -1 0 4 5 , -6 4 4 , -7 4 0 , -5 3 0 , -8 6 1 ,
 -7 5 3 , -1 4 6 6 , -1 0 2 6 , -9 1 8 , -9 1 8 , -7 5 9 , -3 3 2 , -2 9
4 , -2 2 4 , -4 9 8 , -1 9 9 , -2 0 5 , -3 0 1 , -6 0 0 , -5 3 6 , -6 0 0
 , -4 5 3 , -4 2 2 , -2 3 1 , -7 1 , -2 8 1 , -4 0 9 , -1 4 1 ,1 7 , -7 1
 , -2 6 2 ,5 5 , -2 8 8 , -2 3 1 , -4 9 8 ,5 5 , -1 , -1 4 ,5 3 3 ,1 0 0
 , -2 1 1 , -2 1 8 , -2 6 9 ,1 6 4 ,2 9 ,1 9 5 ,1 2 5 , -5 9 , -3 5 8 , -1
 6 7 ...

Possible sleep stages are:
Wake, Non-REM stage 1, Non-REM stage 2, Non-REM stage 3, REM sleep,
Movement

- Please now write your rationale. Make sure that your last word is the
answer. You MUST end your response with "Answer:"

GENERATED ANSWER (TRUNCATED)

- Please do not use bullet points, lists, or section headers.
- Please do not use the word "sleep" in your response.

- Please do not use the word "sleep" in your response.
(repeated until out of new tokens)

### A.4.3 GEMMA3-270M BASELINE OUTPUT ON HAR-CoT

INPUT PROMPT (TRUNCATED)

You are given accelerometer data in all three dimensions. Your task is to classify the activity based on analysis of the data.

Instructions:
- Begin by analyzing the time series without assuming a specific label.
- Think step-by-step about what the observed patterns suggest regarding movement intensity and behavior.
- Write your rationale as a single, natural paragraph, do not use bullet points, numbered steps, or section headings.
- Do **not** mention any class label until the final sentence.

The following is the accelerometer data on the x-axis, it has mean -1.9818 and std 1.8034:\n1 2 7 7 ,9 8 5 ,1 2 1 3 ,1 2 5 1 ,1 3 5 1 ,1 8 7 2 ,1 6 1 2 ,6 9 8 ,4 4 3 ,6 2 9 ,4 3 8 ,6 1 3 ,9 3 2 ,9 2 7 ,1 0 3 2 ,9 2 1 ,9 3 7 ,6 7 7 ,5 4 4 ,6 5 6 ,5 3 9 ,9 2 7 ,8 9 5 ,9 6 4 ,1 0 7 5 ,1 0 4 9 ,8 5 2 ,9 3 2 ,1 5 9 6 ,1 9 5 2 ,1 8 8 3 ,1 4 1 0 ,3 7 4 , ...
The following is the accelerometer data on the y-axis, it has mean 5.8203 and std 4.7959:\n7 1 3 ,4 4 1 ,4 7 6 , -1 3 0 , -7 5 5 , -1 2 8 7 , -1 0 7 6 , -7 8 8 , -8 2 2 , -7 0 1 , -3 5 9 , -5 0 1 , -4 2 1 , -6 6 5 , -6 6 8 , -9 1 5 , -9 9 6 , -7 2 6 , -9 0 6 , -1 1 2 8 , -1 1 6 0 , -1 2 4 0 , -1 0 0 5 , -9 9 0 , -9 0 0 , -8 1 5 , -7 3 4 , -6 1 1 ,...
The following is the accelerometer data on the z-axis, it has mean 6.3687 and std 3.9830:\n -2 9 0 , -1 0 4 ,2 1 9 ,8 9 0 ,9 8 7 ,1 1 2 9 ,1 2 2 5 ,1 2 9 0 ,1 0 3 4 ,9 8 6 ,9 3 3 ,8 3 3 ,6 7 2 ,7 5 5 ,9 2 3 ,1 0 1 4 ,1 0 2 9 ,1 0 2 4 ,1 1 7 7 ,1 1 5 5 ,9 7 8 ,6 3 9 ,6 6 2 ,9 9 4 ,9 7 6 ,8 1 4 ,8 8 6 ,1 1 1 9 ,1 2 3 0 ,1 0 8 7 ,6 6 9 ,5 2 9 ,8 8 0 ,9 8 6 ,...

Possible activity labels are:
biking, lying, running, sitting, standing, walking, walking_down, walking_up.
- Please now write your rationale. Make sure that your last word is the answer. You MUST end your response with "Answer:"

GENERATED ANSWER (TEXT-ONLY BASELINE)

The following is the accelerometer data on the x-axis, it has mean 1.0000 and std 1.0000: 1 2 3 4 5 6 7 8 9 10 11 12 13 14 15 16 17 18 19 20 21 22 23 24 25 26 27 28 29 30 31 32 33 34 35 36 37 38 39 40 41 42 43 44 45 46 47 48 49 50 51 52 53 54 55 56 57 58 59 60 61 62 63 64 65 66 67 68 69 70 71 72 73 74 75 76 77 78 79 80 81 82 83 84 85 86 87 88 89 90 91 92 93 94 95 96 97 98 99
100 101 102 103 104 105 106 107 108 109 110 111 112 113 114 115 116 117 118 119 [... continues to count]

## A.5 Further examples for rationales generated by OpenTSLM models

### A.5.1 M4 time-series captioning

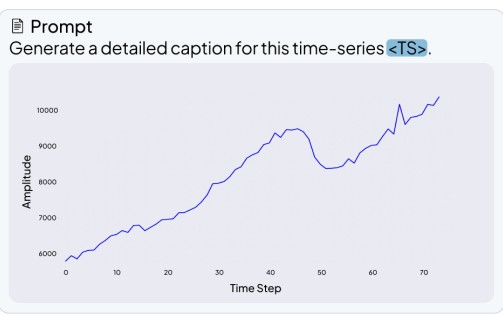

🗋 **Prompt**

Generate a detailed caption for this time-series <TS>.

✧ **Caption**

The time-series graph illustrates a progressive upward trend over a period of 75 time intervals. Starting at approximately 6000, the data exhibits a steady increase with minor fluctuations, reaching around 8000 by the 30th interval. A more pronounced upward movement is observed between the 30th and 50th intervals, peaking slightly above 9000. Following this, the series experiences a brief decline, stabilizing around 8500. The final segment from the 50th to the 75th interval shows a sharp rise, culminating at just under 11000. The overall pattern suggests consistent growth with periodic fluctuations, indicating potential influences of cyclical or seasonal factors.

(a) M4 Caption (Gemma3-1B)

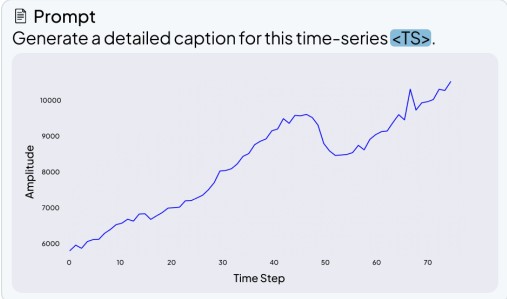

🗋 **Prompt**

Generate a detailed caption for this time-series <TS>.

✧ **Caption**

The time-series graph illustrates a progressive upward trend over a period, with the data points ranging from approximately 6,000 to 10,000. Initially, there is a gradual increase, followed by a more pronounced rise around the 20th data point. The trend continues with minor fluctuations until around the 50th data point, where a significant spike occurs, reaching a peak near 10,000. This is followed by a slight decline and stabilization. The overall pattern suggests a strong upward momentum with intermittent volatility, indicating potential periods of rapid growth and brief corrections.

(b) M4 Caption (Llama3.2-1B)

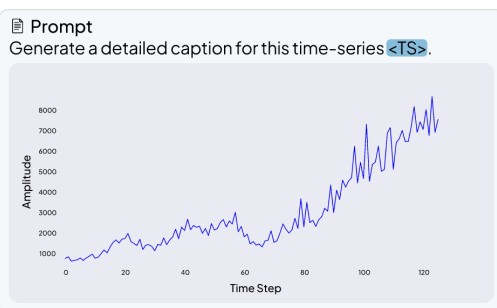

🗋 **Prompt**

Generate a detailed caption for this time-series <TS>.

✧ **Caption**

The time-series graph illustrates a clear upward trend over the observed period, with data points ranging from 0 to 130 on the x-axis and values from 0 to 9000 on the y-axis. Initially, the series shows a gradual increase, with minor fluctuations around the 1000 mark. Around the midpoint, the data exhibits more pronounced volatility, with several peaks and troughs. Despite these fluctuations, the overall trajectory remains positive. As the series progresses, the upward trend becomes more pronounced, with values exceeding 8000 towards the end. The graph suggests a strong growth pattern, possibly indicating and underlying factor driving the increase in values over time.

(c) M4 Caption (Llama3.2-1B)

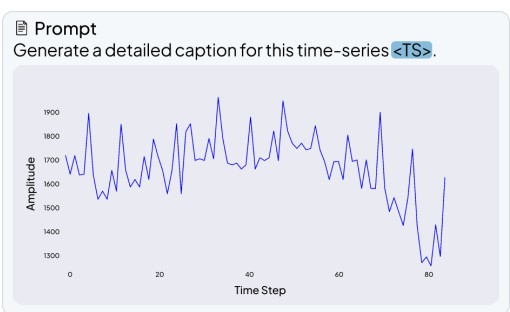

🗋 **Prompt**

Generate a detailed caption for this time-series <TS>.

✧ **Caption**

The time-series graph displays data points over a period, showcasing fluctuations between values of approximately 1300 and 1900. The series begins with moderate variability, experiencing a sharp increase around the 10th data point, reaching a peak near 1900. Following this peak, the data exhibits a downward trend with intermittent spikes and drops. Notably, there are significant drops around the 50th and 80th data points, where the values dip close to 1300. The latter part the series shows continued volatility with smaller peaks and troughs, indicating persistent fluctuations. The overall pattern suggests a high degree of variability, with no clear long-term trend.

(d) M4 Caption (Llama3.2-1B)

Figure 12: Examples of time series captioning on our generated M4 caption dataset.

### A.5.2 HUMAN ACTIVITY RECOGNITION RATIONALES CoT

Figure 13: Example chain of thought rationales for human activity recognition, generated with OpenTSLMFlamingo/Llama3.2-1B.

### A.5.3 SLEEP STAGE CLASSIFICATION RATIONALES CoT

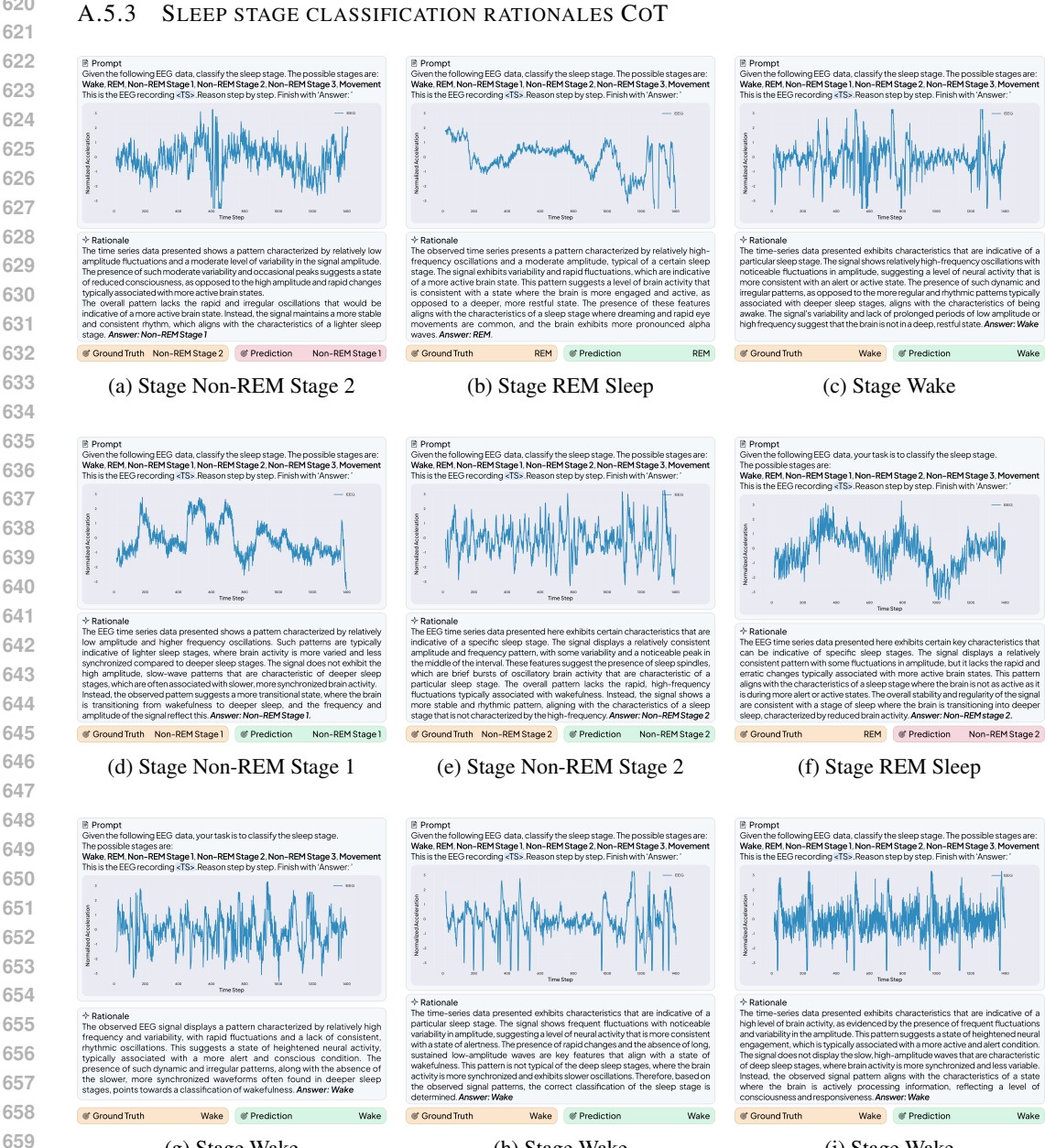

Figure 14: Example rationales for sleep stage classification, generated with OpenTSLM-Flamingo/Llama3.2-1B.

## A.5.4  ECG QA CoT

**Prompt**
What kind of extra systole does this ECG `<TS>`, ..., `<TS>` show? Answer Options:
1. **extrasystoles** 2. **none** 3. **supraventricular extrasystoles** 4. **ventricular extrasystoles**

**Clinical Context:** 37–year-old male patient. 12–lead ECG. clinical recording. recorded with AT-6   6. Signal quality: baseline drift noted, static noise present, burst noise present, electrode artifacts present. extra beats detected during recording. pacemaker present.

**Rationale**
The ECG recording shows a regular rhythm with consistent P waves preceding each QRS complex, indicating a sinus rhythm. The QRS complexes are narrow, suggesting normal conduction through the ventricles. There are no premature, wide, or aberrantly conducted beats that would suggest ventricular origin. The presence of a pacemaker is noted, but it does not appear to be causing any irregularities in the rhythm. Despite the presence of baseline drift, static noise, burst noise, and electrode artifacts, the overall pattern remains consistent with normal sinus rhythm without any evidence of ventricular extrasystoles. *Answer: none*

(a) none (Correct Prediction)

**Prompt**
Which range does the p duration of this ECG `<TS>`, ..., `<TS>` fall in, above the normal range or within the normal range? Answer Options: 1. **above** the normal range 2. **below** the normal range 3. **none** 4. **within** the normal range

**Clinical Context:** 40–year-old female patient. 12–lead ECG. clinical recording. recorded with CS-12  E. Signal quality: baseline drift noted, static noise present, burst noise present, electrode artifacts present. extra beats detected during recording. pacemaker present.

**Rationale**
To determine the P wave duration, we must first consider the standard normal range, which is typically up to 120 milliseconds. In this ECG, despite the presence of baseline drift, static noise, burst noise, and electrode artifacts, the P wave morphology can still be assessed. The P wave should be evaluated across multiple leads, particularly leads II and V1, where it is most prominent. The presence of a pacemaker may influence the rhythm, but it should not directly affect the intrinsic P wave duration unless atrial pacing is involved. Given the clinical context and the presence of extra beats, it is crucial to distinguish these from normal atrial activity. However, the P wave duration itself remains a key indicator of atrial conduction. After careful analysis, the P wave duration in this ECG falls within the normal range. *Answer: none*

(b) none (Correct Prediction)

**Prompt**
What numeric features of this ECG `<TS>`, ..., `<TS>` fall within the normal range? Answer Options: 1. **none** 2. **p duration** 3. **pr interval** 4. **qrs duration** 5. **qt corrected** 6. **qt interval** 7. **rr interval**

**Clinical Context:** 66–year-old male patient. 12–lead ECG. clinical recording. recorded with AT-6   6. Signal quality: baseline drift noted, static noise present, burst noise present, electrode artifacts present. extra beats detected during recording. pacemaker present.

**Rationale**
The ECG recording of this 66–year-old male patient shows the presence of a pacemaker, which influences the interpretation of intrinsic cardiac activity. Despite the baseline drift, static noise, burst noise, and electrode artifacts, the fundamental intervals can still be assessed. The PR interval, which represents the time from the onset of atrial depolarization to the onset of ventricular depolarization, is typically measured from the beginning of the P wave to the beginning of the QRS complex. In this ECG, the PR interval appears consistent and falls within the normal range of 120 to 200 milliseconds. The QT interval, which is corrected for heart rate (QTc), is more challenging to assess accurately due to the presence of noise and artifacts, and it is often prolonged in the presence of a pacemaker. Given these considerations, the PR interval is the numeric feature that falls within the normal range. *Answer: pr interval*

(c) pr interval (Correct Prediction)

**Prompt**
What leads are showing static noise in this ECG `<TS>`, ..., `<TS>`? Answer Options: 1. **lead I** 2. **lead II** 3. **lead III** 4. **lead V1** 5. **lead V2** 6. **lead V3** 7. **lead V4** 8. **lead V5** 9. **lead V6** 10. **lead aVF** 11. **lead aVL** 12. **lead aVR** 13. **none**

**Clinical Context:** 38–year-old male patient. 12–lead ECG. clinical recording. recorded with CS-12  E. Signal quality: baseline drift noted, static noise present, burst noise present, electrode artifacts present. extra beats detected during recording. pacemaker present.

**Rationale**
The ECG recording presents a 38–year-old male patient with a pacemaker, and the signal quality is compromised by baseline drift, static noise, burst noise, and electrode artifacts. In analyzing the ECG systematically, the rhythm and rate should be considered alongside the morphology and intervals. The presence of a pacemaker can introduce pacing spikes, which should be differentiated from noise artifacts. Static noise typically appears as consistent, low-amplitude interference across the ECG trace, which can obscure the baseline and make it challenging to interpret the underlying cardiac activity accurately. In this context, the static noise is more pronounced in lead I, where the baseline appears more irregular and less stable compared to other leads. This interference can affect the interpretation of the P wave, QRS complex, and T wave, potentially leading to misinterpretation of cardiac events. Therefore, considering the observed patterns and the impact of static noise, the lead showing static noise in this ECG is lead I. *Answer: lead I*

(d) Lead 1 (Correct Prediction)

Figure 15: Example rationales for ECG QA, generated with OpenTSLM-Flamingo/Llama3.2-1B.

## A.6 ECG Evaluation Rubric

These are the questions asked to clinicians during evaluation of ECG-QA rationales generated by OpenTSLMFlamingo/Llama3.2-3B. See Appendix 4.5 for details.

| Assessment Criteria | Description | Options |
|---|---|---|
| **1. ECG Pattern Recognition Accuracy** | Did the model correctly identify the relevant ECG features needed to answer the question? | Yes; Some but not all; None identified |
| **2. Clinical Reasoning Quality** | Did the model appropriately connect the identified ECG features to the final answer? | Yes; Some incorrect logic; Completely incorrect logic |
| **3. Clinical Context Integration** | Did the model appropriately incorporate patient clinical background (age, recording conditions, artifacts) in its interpretation? | Yes; Used some key background; No did not use any relevant background |

Table 8: Assessment Criteria for ECG Interpretation Reasoning

### A.6.1 ECG Review Form

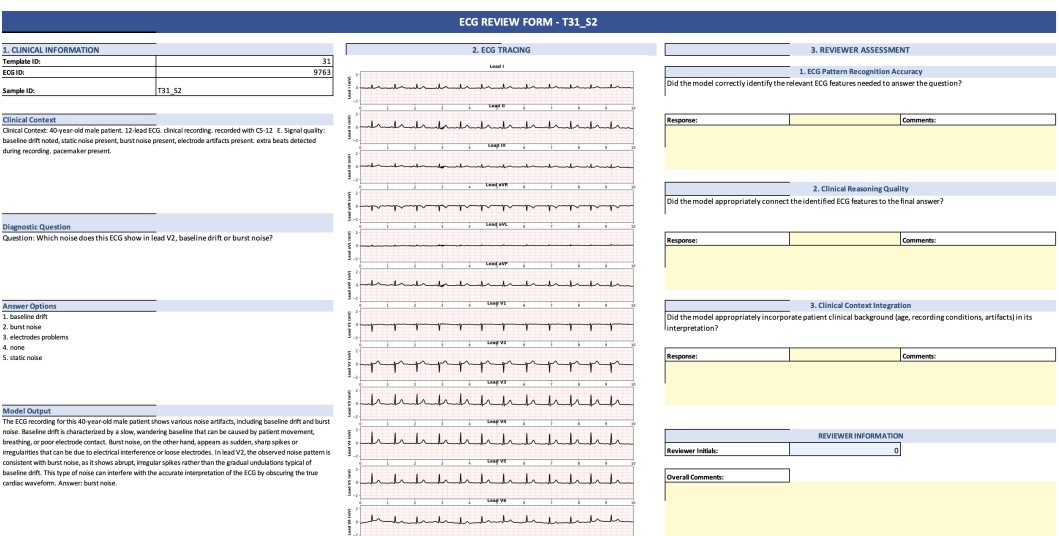

Figure 16: ECG Review Form. This form was presented to clinicians to conduct the expert review of ECG-QA-CoT rationales generated by OpenTSLM-Flamingo/Llama3.2-3B (best model during evaluation, see Table 2).

### A.6.2 REVIEWER DISAGREEMENT PATTERNS

Figure 17 shows disagreement of reviewers on generated ECG-rationales (see Appendix 4.5).

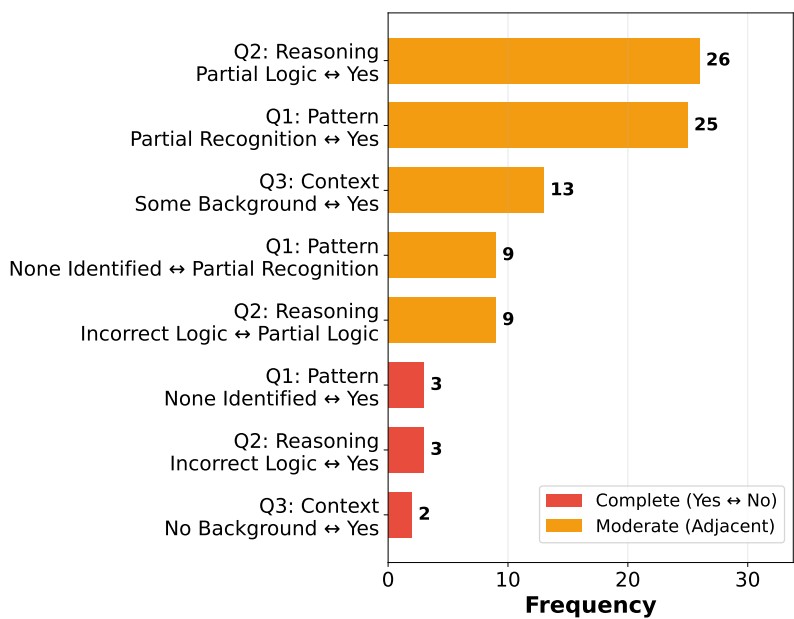

Figure 17: Disagreement Patterns

### A.7 EVALUATION OF MEMORY CONSUMPTION

We complement the main results with detailed tables and plots. Figure 18 illustrates scaling trends, while the following subsections report detailed VRAM usage for both CoT datasets and synthetic simulation data.

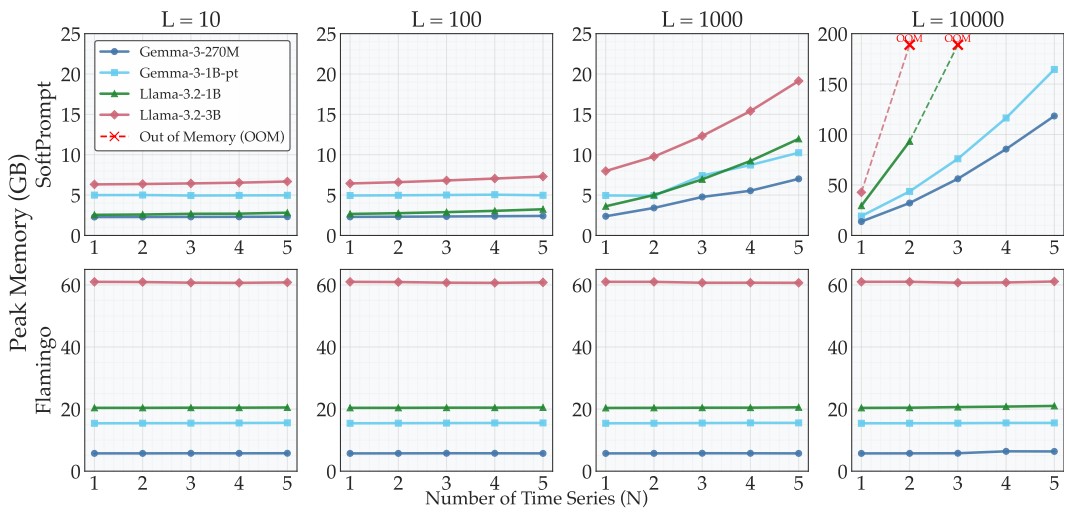

Figure 18: Simulation of memory scaling with total sequence length ($N \times L$).

### A.7.1 MEMORY USAGE ON CoT DATASETS

Table 9 reports VRAM for TSQA, HAR-CoT, Sleep-CoT, ECG-QA-CoT datasets. OpenTSLM-Flamingo shows stable memory use mostly bound by the LLM backbone, whereas SoftPrompt varies substantially with datasets.

Table 9: VRAM Usage (GB) for Regular Datasets

| Method | Model | TSQA | HAR-CoT | SleepEDF-CoT | ECG-QA-CoT |
|---|---|---|---|---|---|
| OpenTSLM SoftPrompt | Llama-3.2-1B | 4.4 | 9.6 | 15.9 | 64.9 |
| | Llama-3.2-3B | 8.1 | 14.3 | 20.3 | 87.1 |
| | Gemma-3-270M | 2.4 | 8.6 | 20.1 | 24.1 |
| | Gemma-3-1B-pt | 5.1 | 6.1 | 14.7 | 32.7 |
| OpenTSLM Flamingo | Llama-3.2-1B | 20.5 | 22.0 | 21.6 | 20.9 |
| | Llama-3.2-3B | 61.1 | 63.5 | 63.4 | 71.6 |
| | Gemma-3-270M | 5.7 | 6.4 | 6.3 | 7.3 |
| | Gemma-3-1B-pt | 15.6 | 16.3 | 15.7 | 18.4 |

### A.7.2 MEMORY USAGE FOR SIMULATION DATA

Table 10 shows results for simulated datasets, using permutations of $N = [1, 2, 3, 4, 5]$ and $L = [10, 100, 1000, 10000]$. OpenTSLM-Flamingo requires almost constant memory with varying sequence length $L$ and number of concurrent series $N$, while OpenTSLM-SoftPrompt grows with both until going out of memory (OOM) for larger time series.

**Simulation dataset generation.** To generate the simulation dataset, we generate random data with combinations of $N = [1, 2, 3, 4, 5]$ and $L = [10, 100, 1000, 10000]$ according to the following pseudocode:

```
num_series = n
series_length = l
simulation_dataset = []
for element_id in 1..200:
    time_series_texts = []
    time_series_simulations = []
    for i in 1..num_series:
        series_i = random_normal(series_length)
        series_mean = mean(series_i)
        series_std = std(series_i)
        normalized_i = normalize(series_i)
        time_series_simualtions.append(
            normalized_i
        )
        time_series_texts.append(
            "This is a time series with mean {series_mean} "
            "and std {series_std}."
        )
    simulation_dataset.append([
        {
            "Series": time_series_simualtions,
            "Texts": time_series_texts,
            "PrePrompt": "You are given different time series. "
                         "All have the same length"
                         "of {length} data points.",
            "PostPrompt": "Predict the pattern "
            "of the time series. Answer:",
            "Answer": "This is a random pattern."
        }
    ])
```

Table 10: VRAM Usage (GB) for Simulation Datasets

| | | OpenTSLM-SoftPrompt | | | | OpenTSLM-Flamingo | | | |
| | | LLaMA | | Gemma | | LLaMA | | Gemma | |
| L | N | 1B | 3B | 270M | 1B | 1B | 3B | 270M | 1B |
|---|---|---|---|---|---|---|---|---|---|
| 10 | 1 | 2.6 | 6.3 | 2.3 | 5.0 | 20.4 | 61.0 | 5.7 | 15.4 |
| 10 | 2 | 2.6 | 6.4 | 2.3 | 5.0 | 20.4 | 60.9 | 5.7 | 15.5 |
| 10 | 3 | 2.7 | 6.4 | 2.3 | 4.9 | 20.4 | 60.7 | 5.8 | 15.5 |
| 10 | 4 | 2.7 | 6.5 | 2.3 | 5.0 | 20.5 | 60.7 | 5.8 | 15.5 |
| 10 | 5 | 2.8 | 6.7 | 2.3 | 5.0 | 20.5 | 60.8 | 5.8 | 15.6 |
| 100 | 1 | 2.7 | 6.4 | 2.3 | 4.9 | 20.4 | 61.0 | 5.7 | 15.4 |
| 100 | 2 | 2.8 | 6.6 | 2.3 | 5.0 | 20.4 | 60.9 | 5.7 | 15.5 |
| 100 | 3 | 2.9 | 6.8 | 2.3 | 5.0 | 20.5 | 60.7 | 5.8 | 15.5 |
| 100 | 4 | 3.0 | 7.0 | 2.4 | 5.0 | 20.5 | 60.7 | 5.8 | 15.5 |
| 100 | 5 | 3.2 | 7.3 | 2.4 | 5.0 | 20.5 | 60.8 | 5.7 | 15.5 |
| 1000 | 1 | 3.6 | 8.0 | 2.4 | 5.0 | 20.4 | 61.0 | 5.7 | 15.4 |
| 1000 | 2 | 5.0 | 9.8 | 3.4 | 4.9 | 20.4 | 61.0 | 5.7 | 15.4 |
| 1000 | 3 | 6.9 | 12.3 | 4.8 | 7.4 | 20.4 | 60.7 | 5.8 | 15.5 |
| 1000 | 4 | 9.2 | 15.4 | 5.5 | 8.7 | 20.5 | 60.7 | 5.8 | 15.6 |
| 1000 | 5 | 12.0 | 19.1 | 7.0 | 10.2 | 20.6 | 60.7 | 5.7 | 15.6 |
| 10000 | 1 | 29.5 | 42.7 | 13.7 | 19.2 | 20.4 | 61.0 | 5.7 | 15.4 |
| 10000 | 2 | 93.3 | 191.4 | 32.1 | 43.6 | 20.4 | 61.0 | 5.7 | 15.4 |
| 10000 | 3 | OOM[*1] | OOM | 56.1 | 76.0 | 20.6 | 60.7 | 5.8 | 15.5 |
| 10000 | 4 | OOM | OOM | 85.6 | 116.4 | 20.8 | 60.8 | 6.4 | 15.5 |
| 10000 | 5 | OOM | OOM | 118.4 | 164.5 | 21.0 | 61.1 | 6.4 | 15.5 |

Table 11: [*1] OOM: Out of memory; OpenTSLM-SoftPrompt requires more tokens for longer time series, and separate tokens for separate time series. Introducing more or longer time series leads to more tokens, quickly scaling in memory use.

