# OpenReview forum: "OpenTSLM: Time-Series Language Models for Reasoning over Multivariate Medical Text- and Time-Series Data"
_ICLR.cc/2026/Conference — Submitted to ICLR 2026_

### Official Review · Reviewer_M85T · 2025-10-30

**Soundness:** 2
**Presentation:** 3
**Contribution:** 2
**Rating:** 4
**Confidence:** 4

**Summary:**

The work introduces two architectures for medical time series and text multi-modal tasks.

**Strengths:**

1. The methodology section on the model architecture was quite clear to follow.
2. Including expert evaluation on model responses is valuable and provides useful insight into the model’s reasoning capability.

**Weaknesses:**

1. The authors propose two model architectures but only compare against vanilla finetuned LLM baselines. Additional baselines would strengthen the study:
-  Since the architectures repurpose VLM-style designs, it would be useful to test encoding time series as images and see how a fine-tuned VLM on the same dataset split compare to explicit time series modeling.
- Existing time series–text alignment methods such as [1](https://arxiv.org/pdf/2412.03104), [2](https://arxiv.org/pdf/2408.07773) should be considered for comparison.
- For repurposed tasks like ECG-QA, it would make sense to compare against task specific models, even if they are not multimodal. If adding textual components does not improve results, the benefit of enabling reasoning becomes unclear.

2. The paper presents two frameworks but does not analyze the motivation behind keeping both. The abstract hypothesizes that Flamingo would outperform SoftPrompt, yet there is no follow-up analysis of this hypothesis, and Table 2 shows SoftPrompt performing better on most datasets.

3. The experimental results would benefit from more fine-grained analysis. For example, in Table 2, LLaMA-1B consistently outperforms LLaMA-3B with softprompt, which is counter-intuitive. It would also help to show how performance scales with model size to understand whether the method is scalable.

4. The three CoT datasets were annotated using ChatGPT, with only limited manual review mentioned. To increase reliability, it would be helpful to provide dataset review statistics, reviewers’ backgrounds, and potentially include studies such as human detectability or deliverability checks for future community use.

**Questions:**

Please see weaknesses

---

> ### Author Response · Authors · 2025-11-25
> **Addressing reviewer weaknesses - W1**
>
> Thank you for your feedback, we appreciate the scientific discourse. Please find below our discussion on the proposed weaknesses.
>
> ### **W1: Comparing two model architectures**
>
> - **Since the architectures repurpose VLM-style designs, it would be useful to test encoding time series as images and see how a fine-tuned VLM on the same dataset split compare to explicit time series modeling.**
>
> Thank you for raising this point. We agree that comparing fine-tuned VLMs makes sense, and we are currently running an evaluation on this.
>
> Additionally, we would like to clarify the motivation for our architectural choices. Although our model draws inspiration from VLM architectures, time series is a fundamentally different modality. Our use of Flamingo-style components is based on their structural properties, specifically, the connector design (Perceiver Resampler, Gated Cross-Attention) that enables effective fusion of multi-modal or multi-resolution inputs by incorporating external features upon encountering special tokens in the text (image and endofchunk) rather than on any visual-processing aspect. We adapt these mechanisms to the characteristics of time-series data, not images. Our work adopts Flamingo’s structure for multimodal-fusion to time-series modeling, but we do not retain any vision-encoding related parts.  In any case, we appreciate the suggestion and are including such experiments for completeness.
>
> - **Existing time series–text alignment methods such as 1, 2 should be considered for comparison.**.
> Thank you for pointing out the additional literature. We included both in the related work section.
>
> - **For repurposed tasks like ECG-QA, it would make sense to compare against task specific models, even if they are not multimodal. If adding textual components does not improve results, the benefit of enabling reasoning becomes unclear**
>
> Thank you for raising this point. While we do acknowledge that classical, unimodal models are a good baseline for comparison,  we want to highlight why we believe that multimodal, LLM-based reasoning remains valuable even when accuracy is similar or worse. Reasoning is not merely a mechanism for improving predictive performance, it is an output modality with practical utility. A task-specific ECG classifier typically produces a label (e.g., Arrhythmia: 99%). In contrast, our architecture can provide a structured explanation (e.g., “irregular R–R intervals indicative of atrial fibrillation…”). In settings such as healthcare, the ability to produce interpretable, text-based justifications is often as important as marginal improvements in accuracy. A slightly more accurate black-box model is not necessarily more useful than one that communicates its diagnostic rationale to clinicians.
>
> A secondary benefit is thatmultimodal reasoning provides capabilities that conventional ECG models cannot match. LLM-based reasoning enables access to domain knowledge that lies outside the time-series signal itself. Conventional ECG classifiers operate solely on the waveform. Our Flamingo-style architecture can combine the ECG with contextual variables and the LLM’s encoded medical knowledge. For example, queries such as “Given this ECG and an 85-year-old patient, what is the risk level of..?” require integrating signal information with medical guidelines. Something traditional ECG models cannot do. This represents a capability gap, not simply an accuracy difference.
>
> In addition, it is important to clarify that post-hoc explanations generated by pairing a unimodal classifier with a separate LLM do not resolve this limitation. Such explanations are not grounded in the classifier’s internal representations, and there is no mechanism to ensure consistency between the predicted label and the generated rationale. Our model, by conditioning both prediction and explanation on the same fused representation, supports more coherent and faithful reasoning.
>
> A third point concerns generality. While one can always build a tailored model for each specific time-series task, this approach does not scale to the diversity of real-world applications. A multimodal LLM that can answer questions or follow instructions about any time series using natural language provides a significantly broader and more flexible capability. This mirrors the evolution in vision: task-specific image classifiers remain strong baselines, but vision–language models have become far more versatile and widely adopted because they unify classification, captioning, VQA, and retrieval under a single interface. Similarly, a general time-series–language model can serve as a unified backbone for disparate tasks, reducing the need for bespoke architectures while enabling richer, instruction-driven interaction with time-series data.

---

> > ### Author Response · Authors · 2025-11-25
> > **Addressing reviewer weaknesses - W2**
> >
> > ### **W2: The paper presents two frameworks but does not analyze the motivation behind keeping both. The abstract hypothesizes that Flamingo would outperform SoftPrompt, yet there is no follow-up analysis of this hypothesis, and Table 2 shows SoftPrompt performing better on most datasets**
> >
> > We thank the reviewer for raising this concern. Our motivation for including both approaches is that they represent two complementary scalability regimes, and analyzing them together enables a scalability-dependent study of how different architectural choices behave as time-series complexity increases. While we acknowledge SoftPrompt’s better performance on some of the current benchmarks, we argue that the Flamingo-style architecture provides a critical advantage in scalability that SoftPrompt lacks: decoupling sequence length from context usage. SoftPrompt maps time-series patches directly to input tokens. As the resolution or duration of the time series increases (e.g., high-frequency ECG or long-history financial data), the number of tokens grows linearly. This rapidly saturates the LLM's context window, increasing quadratic attention costs and limiting the space available for textual reasoning or few-shot examples. The Flamingo architecture utilizes a Perceiver Resampler to compress variable-length temporal features into a fixed number of visual/temporal tokens before cross-attention. This means we can process a time series of 500 points or 50,000 points using the same number of query vectors in the LLM. Flamingo is architecturally more future-proof for long-sequence modeling. SoftPrompt is a strong baseline for short sequences, but Flamingo offers the scalability required for high-resolution, real-world time-series tasks where context management is critical.

---

> > > ### Author Response · Authors · 2025-11-25
> > > **Addressing reviewer weaknesses - W3**
> > >
> > > ### **W3: The experimental results would benefit from more fine-grained analysis. For example, in Table 2, LLaMA-1B consistently outperforms LLaMA-3B with softprompt, which is counter-intuitive. It would also help to show how performance scales with model size to understand whether the method is scalable**
> > >
> > > We appreciate the reviewer’s suggestion to include more fine-grained analysis and agree that the inverse scaling trend observed in Table 2 (LLaMA-1B > LLaMA-3B with soft prompts) warrants explicit discussion. We emphasizeclarify that this behavior does not indicate a lack of scalability in the proposed method; rather, it reflects the data-constrained setting of our experiments.
> > >
> > > We argue that the 3B model underperforms primarily because pairing a larger backbone with LoRA fine-tuning on relatively small time-series datasets increases susceptibility to overfitting. The 3B model has far greater capacity to memorize the few-shot patterns available, while the 1B model, being more parameter-constrained, is compelled to learn more generalizable representations. Additionally, larger language models carry stronger pre-trained priors. Redirecting a 3B model toward understanding numerical time series, a modality not emphasized during pre-training, using limited supervision, is inherently more difficult. In contrast, the 1B model is more “plastic” and easier to adapt to small amounts of data.
> > >
> > > More broadly, scalability should be interpreted through the lens of the data-to-model ratio. Our results suggest that the method does scale, but only when increases in model size are matched by corresponding increases in available training data. Under the current benchmark conditions, the datasets are simply too small to leverage the full capacity of a 3B model, leading to the observed saturation effect. We will revise the manuscript to articulate this point more clearly and to include additional analysis of performance as a function of model size.

---

> > > > ### Author Response · Authors · 2025-11-25
> > > > **Addressing reviewer weaknesses - W4**
> > > >
> > > > ### **W4: The three CoT datasets were annotated using ChatGPT, with only limited manual review mentioned. To increase reliability, it would be helpful to provide dataset review statistics, reviewers’ backgrounds, and potentially include studies such as human detectability or deliverability checks for future community use**
> > > >
> > > >
> > > > Thank you for highlighting this important issue. We fully agree that thorough validation of the CoT-annotated datasets is essential for ensuring reliability and for supporting future community use. While we would ideally provide 100% curated datasets, several practical constraints limited the extent of manual review in this work. Some datasets, such as ECG-QA with over 240k samples, are prohibitively large for full expert annotation, and additionally, some datasets (e.g., sleep staging, ECG interpretation) require domain specialists whose time is limited. This challenge is not unique to our work and, consistent with prior work on multimodal CoT generation, large-scale expert annotation is often infeasible. As a result, scalable synthetic annotation is necessary to create training corpora of sufficient size.
> > > >
> > > > Importantly, the primary goal of this paper is not to release a definitive, fully curated dataset. Our objective is to evaluate architectural choices for TSLMs. For this purpose, the datasets serve as a practical and reproducible training resource. Our experiments demonstrate that, even with limited manual review, the resulting models substantially outperform strong baselines, including fine-tuned tokenized models and GPT-4o with image inputs. This indicates that the dataset quality is sufficient for the intended scientific objective of analyzing TSLM architectures.
> > > >
> > > > That said, we acknowledge the value of more systematic quality assessments. We also agree that human detectability and deliverability checks would meaningfully strengthen the datasets' utility, and we view these as important directions for follow-up work. We have already initiated collaborations with medical experts for expanded validation in future studies.

---

### Official Review · Reviewer_pnCZ · 2025-10-31

**Soundness:** 3
**Presentation:** 4
**Contribution:** 2
**Rating:** 4
**Confidence:** 4

**Summary:**

This paper presents OpenTSLM, a family of time-series language models that integrate time-series data as a native modality into large language models for reasoning over multivariate medical text-and-time-series data. The work proposes two architectural variants: OpenTSLM-SoftPrompt, which directly maps time-series segments to learnable tokens, and OpenTSLM-Flamingo, which uses a Perceiver Resampler to compress sequences into fixed-size latent representations. The paper demonstrates strong performance on medical tasks including sleep staging, activity recognition, and ECG interpretation.

The paper is well-organized and clearly written, with high-quality figures and substantial experimental work in the medical domain. However, several significant limitations impact the contribution: insufficient coverage of related work on time-series-text multimodal fusion, questions about the training methodology for chain-of-thought reasoning, narrow evaluation metrics that focus only on accuracy without assessing generation quality, and potential concerns about the data collection methodology.

**Strengths:**

1. **Clear presentation and organization**: The paper is well-structured and easy to follow, with high-quality figures that effectively illustrate the architectures and experimental results.
2. **Comprehensive medical domain evaluation**: The authors conduct extensive experiments across multiple medical tasks (sleep staging, activity recognition, ECG interpretation) with clinical validation, demonstrating practical relevance.
3. **Dual architecture design**: Providing both SoftPrompt and Flamingo variants addresses different computational constraints and sequence length requirements, offering flexibility for different use cases.
4. **Strong empirical results**: The models achieve competitive performance on medical tasks, with notable improvements over baseline approaches.
5. **Open-source contribution**: Releasing code, datasets, and model weights promotes reproducibility and future research.

**Weaknesses:**

1. **Insufficient coverage of related work**: The paper lacks a systematic discussion of existing approaches to time-series-text multimodal fusion. Several relevant works that address similar problems are missing or inadequately discussed:

   - **ITFormer** and similar models that bridge time series and natural language
   - **Time-VLM** and other vision-language models adapted for time-series
   - Recent work on addressing modality gap in time-series-text fusion

   The paper does not clearly articulate how OpenTSLM differs from or advances beyond these existing solutions. Given that the problem of modality gap in time-series-text fusion has been addressed through various approaches, the technical contribution needs better positioning relative to this body of work.
2. **Training methodology for chain-of-thought reasoning**: The paper uses supervised fine-tuning (SFT) for training chain-of-thought (CoT) reasoning capabilities. However, there are open questions about whether SFT alone is sufficient:

   - CoT reasoning often requires learning complex reasoning patterns that may benefit from reinforcement learning (RL) approaches, as demonstrated in recent LLM training (e.g., RLHF, RLAIF)
   - The paper does not provide ablation studies comparing SFT-only training with methods incorporating RL
   - No discussion of why RL was not considered or whether it would improve reasoning quality

   This is particularly important for medical applications where reasoning quality and interpretability are critical.
3. **Limited evaluation metrics**: The evaluation focuses primarily on accuracy for structured outputs (multiple-choice questions) but provides limited assessment of the quality of generated text, especially for chain-of-thought reasoning:

   - No metrics for evaluating the quality of generated CoT explanations (fluency, coherence, medical correctness)
   - No human evaluation of explanation quality
   - No automated metrics for text generation (BLEU, ROUGE, semantic similarity, medical knowledge correctness)
   - The paper emphasizes "human-interpretable reasoning outputs" but does not evaluate interpretability

   For a paper focused on reasoning and explanation generation, this is a significant gap.
4. **Data collection methodology concerns**: The paper mentions using images to generate captions and chain-of-thought reasoning for multimodal models, but several questions arise:

   - Is it appropriate to use image-to-text generation for creating training data for time-series models? This seems like a domain mismatch
   - How do captions and CoT explanations generated from images transfer to time-series data?
   - No ablation or analysis showing the impact of this data collection strategy
   - Potential concerns about data quality and relevance
5. **Missing technical details**:

   - Insufficient analysis of when to use SoftPrompt vs. Flamingo (decision criteria)
   - Limited ablation studies on key components (Perceiver Resampler compression ratio, cross-attention mechanisms)
   - No analysis of information loss in the Flamingo compression process

**Questions:**

* **Related work positioning**: Please provide a systematic comparison with existing time-series-text multimodal approaches, particularly ITFormer, Time-VLM, and other recent works. How does OpenTSLM's contribution differ from or build upon these methods? What specific limitations do existing approaches have that OpenTSLM addresses?
* **Training methodology**: Why was reinforcement learning not considered for training chain-of-thought reasoning? Recent work has shown that RL can significantly improve reasoning quality in LLMs. Please provide:
  - Ablation study comparing SFT-only vs. SFT+RL training
  - Discussion of trade-offs and rationale for the chosen approach
  - Analysis of reasoning quality with current training method
* **Evaluation of generation quality**: The paper should include comprehensive evaluation of generated CoT explanations:
  - Add human evaluation of explanation quality (medical correctness, clarity, completeness)
  - Include automated metrics: BLEU, ROUGE, semantic similarity, domain-specific metrics for medical reasoning
  - Compare generated explanations with expert-written ones
  - Assess the correlation between explanation quality and task performance
* **Data collection methodology**: Please clarify and justify the approach of using images to generate captions and CoT for time-series training data:
  - Provide ablation study showing the contribution of image-generated data
  - Analyze potential domain mismatch issues
  - Consider alternatives such as expert-written explanations or synthetic time-series-specific generation
  - Discuss data quality and filtering processes
* **Architecture selection criteria**: Provide clear guidance on when to use SoftPrompt vs. Flamingo:
  - Decision criteria based on sequence length, available resources, task type
  - Comparative analysis of trade-offs
  - Recommendation framework for practitioners
* **Technical ablations**: Include ablation studies on:
  - Perceiver Resampler compression ratio selection (why 64×N?)
  - Impact of different cross-attention mechanisms
  - Information loss analysis in the compression process

---

> ### Author Response · Authors · 2025-11-25
> **Answers to reviewer questions, Q1**
>
> Thank you very much for your review and insightful comments and for recognizing the strengths of OpenTSLM. We appreciate the scientific debate. Please find below the answers to your questions (based on the proposed Weaknesses).
>
> ### **Q1: Related work positioning**
> We thank the reviewer for this observation. We will expand our comparison with prior work in the related works and discussion section. Additionally, below we provide a detailed comparison of our work compared to existing methods (addressing both comments 1 and 2 above).
>
> **1. Architectural contributions compared to prior work**
> We find that the closest existing work is ITFormer, and the work proposed by Chow et al. Both methods leverage softprompting, i.e., early fusion of text and time-series by concatenating text tokens with tokens that encode time series information derived from a trainable encoder. While Chow et al. use a simple PatchTST-based encoder to encode time series into tokens for LLMs, ITFormer uses a more sophisticated time-series encoder that leverages additional positional encodings (to encode time series, channel, and segment positions) and a specific attention mechanism to generate bespoke tokens from the time series data. ITFormer encodes simultaneous readings from (N) sensors encoded into a sequence of tokens that are fed directly into a frozen LLM, i.e., early fusion at the token level. While highly relevant, this setup has two shortcomings for our target setting. First, ITFormer is designed and evaluated for a single multivariate time series (engine sensors) per query, and does not support mixing heterogeneous, unrelated time series in a flexible prompt format. In contrast, OpenTSLM supports prompts that interleave arbitrary numbers and types of time series with text (e.g., “here are the heart-rate readings over the last 7 days”, “here are activity and step-count patterns from the last 4 weeks”, plus “here are the patients subjective symptom scores over the 24 hours”) via our chunked prompt format and special ⟨TS⟩ / ⟨endofchunk⟩ tokens.
> Second, ITFormer’s early-fusion design encodes time series as additional LLM tokens, so memory and compute scale quadratically with the combined (text + tokenized time series) sequence length. We show empirically that such soft-prompt/early-fusion approaches become impractical for long and multi-lead clinical signals (see Appendix A.7). OpenTSLM-Flamingo instead keeps time-series latents in a separate stream and fuses them via gated cross-attention layers. This decouples LLM context length from time-series length, leading to VRAM usage being bounded by the dimension of the embeddings produced by our time series encoder, as we increase sequence length or number of series (e.g., Llama-1B stays around 20–22 GB; Llama-3B around 61-72 GB across TSQA, HAR-CoT, Sleep-CoT, and ECG-QA-CoT), while maintaining or improving performance on long ECG-QA sequences. Methodologically, OpenTSLM-Flamingo leverages late-fusion via cross-attention, and ITFormer leverages early fusion via time-series encoded as LLM tokens.
>
> Other methods such as MedualTime, Time2Lang, SensorLLM, and MedTsLLM typically target specific tasks (e.g., ECG classification on PTB-XL, forecasting, activity recognition) and rely on fixed classification or regression heads (projection layers), removing free-form text generation capabilities. In contrast, OpenTSLM is explicitly trained as a time-series language model: the same model architecture is used for TSQA, chain-of-thought classification (HAR-CoT, Sleep-CoT), ECG question answering, and time-series captioning, outputting answers, classifications or captions in natural language.
>
> **2. Systematic comparison of soft prompting vs. cross-attention for multimodal modelling**
> Existing TS–LLM work typically adopts a single fusion strategy, either soft prompting (e.g., Chow et al., MedualTime, Time2Lang) or cross-attention (e.g., SensorLM-like designs). OpenTSLM is, to our knowledge, the first to implement and directly compare both paradigms within the same framework (OpenTSLM-SoftPrompt vs. OpenTSLM-Flamingo), showing that cross-attention yields stable memory and performance for long, multivariate clinical signals where soft prompting becomes impractical (e.g., 80k-token ECG sequences). This provides concrete guidance for future TSLM design rather than proposing another single-architecture variant.
>
> **3. Expert evaluation of CoT rationales on ECG-QA**
> OpenTSLM contributes three new CoT datasets (HAR-CoT, Sleep-CoT, ECG-QA-CoT) built from real medical and wearable data and trains models to produce chain-of-thought rationales, not just labels. We further perform an expert study with cardiologists on ECG-QA-CoT, who rate 92.9% of our ECG interpretations as correct or partially correct, demonstrating that OpenTSLM’s reasoning is clinically meaningful and not merely a numerical improvement.

---

> > ### Author Response · Authors · 2025-11-25
> > **Answers to reviewer questions [continued] - Q2**
> >
> > ### **Q2: Training methodology - Reasoning, Reinforcement Learning**
> > We appreciate the reviewer’s insightful suggestion. While RL is a promising direction for enhancing reasoning quality, our current work intentionally focuses on architectural contributions rather than optimization techniques. We do not include RL at this stage because doing so would introduce several confounding factors that make it difficult to isolate and evaluate the benefits of the proposed architecture itself. Also, incorporating RL would require substantial additional engineering, such as designing reliable reward functions for time-series interpretation and collecting expert-labeled preference data, and would complicate attribution of improvements between architecture and training method. Our goal in this paper is to demonstrate the advantages of the proposed TSLM architectures. Nevertheless, we fully agree that RL-based refinement is a valuable next step, particularly given recent progress in RLHF for CoT generation, and we view it as a natural direction for future work to further strengthen the reasoning quality of TSLMs. Below, we elaborate on why RL is out of scope for the present work.
> >
> > **First**, our central goal is to introduce TSLMs as a new architectural class for multimodal LLMs that can handle text and raw multivariate time-series data. Incorporating RL would add an additional optimization layer, making it difficult to attribute performance gains to the architecture versus the training algorithm. We therefore opted for a clean evaluation based purely on supervised finetuning (cross-entropy SFT) using ground-truth CoT traces. This design isolates and highlights the contribution of time-series modeling mechanisms (soft prompting vs. cross-attention).
> >
> > **Second**, RL in time-series settings requires a reliable reward function. Unlike mathematical or symbolic reasoning, tasks such as ECG interpretation, sleep staging, and activity recognition involve subtle temporal patterns, clinical nuance, and expert-dependent interpretations. Developing reward models for physiological reasoning (e.g., clinical-correctness rewards, hallucination penalties, or feature-detection-–based rules) would itself require a large, expert-labeled corpus and careful safety checks, —resources beyond the scope of this initial architectural investigation.
> >
> > **Third**, cross-entropy SFT already gives strong performance. Our results show that even without RL, OpenTSLM models substantially outperform all baselines, —including GPT-4o with image inputs and tokenized-finetuned LLMs, —across HAR, SleepEDF, and ECG-QA. Moreover, our expert evaluation on ECG-QA demonstrates that SFT alone produces clinically meaningful reasoning: cardiologists rated 92.9% of sampled rationales as correct or partially correct (Appendix A.5)
> > Value of RL and outlook for future work.
> > We view RLHF as a valuable next step for TSLMs. RL could encourage phrasing-invariant, clinically grounded reasoning and allow the model to optimize for correctness and explanation quality. Designing reliable reward functions, potentially combining answer-based rewards, rule-based signal detectors (e.g., QRS detection, sleep spindles), or rewards for medically sound reasoning, is an exciting area for future work. Our current supervised results establish a strong architectural foundation, and we see RL-based refinement as a promising direction for further enhancing reasoning quality in TSLMs.

---

> > > ### Author Response · Authors · 2025-11-25
> > > **Answers to reviewer questions [continued] - Q3**
> > >
> > > ### **Q3: Evaluation of generation quality**
> > > Thank you very much for raising this important point. We fully agree that a comprehensive evaluation of generated chain-of-thought (CoT) explanations is essential, especially in clinical settings where correctness, clarity, and reasoning quality matter as much as accuracy. Our approach in this work was shaped by the need to prioritize evaluation methods that reflect real clinical utility. For this reason, we focused primarily on expert-based clinical evaluation, which we consider the most meaningful indicator of explanation quality in high-stakes medical contexts. Automated metrics such as BLEU or ROUGE are limited in their ability to capture medical correctness, causal reasoning, or waveform-feature grounding, and therefore offer an incomplete or sometimes misleading picture. This informed our choice to center the evaluation on expert review of interpretability and clinical validity. We would like to direct the reviewer to Sections 4.5 and Appendix A.6, where this evaluation is described, and we are happy to revise the manuscript to make this emphasis more explicit. We also welcome further guidance on which additional aspects of explanation quality the reviewer believes could still be strengthened.
> > > To elaborate further, this expert evaluation was conducted on the ECG-QA dataset by five clinicians, using a rubric grounded in the American College of Cardiology/American Heart Association Clinical Competence Statement and the RIME (Reporter-Interpreter-Manager-Educator) framework. The review examined:
> > >
> > > 1. **ECG Feature Recognition:**
> > > Whether the model correctly identified relevant waveform features.
> > > 2. **Clinical Reasoning:**
> > > Whether the model appropriately connected these features to its diagnostic conclusion, reflecting coherent and medically grounded reasoning rather than pattern matching.
> > > 3. **Clinical Context Integration:**
> > > Whether the rationale incorporated relevant patient-specific information (e.g., age, artifacts, acquisition details).
> > >
> > >
> > > Across these, the model produced correct or partially correct rationales in 92.9% of cases as assessed by the human expert reviewers, demonstrating strong alignment with expert expectations for coherent and clinically valid reasoning.
> > > For transparency, we provide the following is the exact excerpt from the originally submitted manuscript (see Section 4.5, with additional details in Appendix A.6) :
> > > “To evaluate the quality of model rationales, we conducted an expert review with five cardiologists from ANONYMIZED Hospital on rationales generated by OpenTSLM-Flamingo-Llama3.2-3B (best model) for ECG-QA.
> > > Evaluation followed a rubric derived from the American College of Cardiology/American Heart Association Clinical Competence Statement on ECG  and based on the RIME (“Reporter–Interpreter–Manager–Educator”) framework (see Appendix A.6), assessing whether the model:
> > > (1) correctly identified relevant ECG features;
> > > (2) appropriately connected them to the final answer;
> > > (3) incorporated patient context (age, artifacts, ...).
> > > Overall, the model gave a correct or partially correct ECG interpretation in 92.9\% of cases, spanning ECG recognition, reasoning, and contextualization.”

---

> > > > ### Author Response · Authors · 2025-11-25
> > > > **Answers to reviewer questions [continued] - Q4**
> > > >
> > > > ### **Q4: Data collection methodology**
> > > >
> > > > Thank you very much for raising this point. Consistent with prior work (e.g., Chow et al.), we generate CoT datasets using GPT with time-series plots because no suitable text-time-series CoT datasets exist. We outline further details below.
> > > >
> > > > A major challenge in building Time-Series Language Models (TSLMs) is the absence of existing datasets that pair raw time-series signals with high-quality natural-language rationales. Although several large collections of raw physiological data exist (e.g., SleepEDF, PTB-XL/ECG-QA, HHAR), none provide paired multimodal reasoning traces suitable for training LLMs. To address this, we adopt an approach inspired by Chow et al. (2024), where LLMs are prompted with simple plots of the underlying time series and asked to produce CoT explanations conditioned on both the plot and the ground-truth label (Appendix A.2). This has two practical advantages:
> > > >
> > > > 1. **Scalability**: Datasets such as ECG-QA contain >240k samples, many requiring domain expertise for annotation (e.g., cardiac rhythm, ECG or sleep pattern interpretation), making full annotation and curation very challenging.
> > > >
> > > > 2. **Grounding**: Providing plots ensures the supervising LLM conditions its CoT on visible temporal structure rather than hallucinating explanations from labels alone.
> > > >
> > > > While we would like to provide 100% curated datasets, several practical constraints limited the extent of manual review in this work. Some datasets, such as ECG-QA with over 240k samples, are prohibitively large for full expert annotation, and additionally, some datasets (e.g., sleep staging, ECG interpretation) require domain specialists whose time is limited and we therefore could not manage to manually annotate all existing datasets. Importantly, the primary scientific objective of our paper is not to produce a perfectly curated CoT corpus, but to demonstrate that multimodal LLMs for time series, i.e., TSLMs, can natively reason over raw time-series inputs.
> > > > Regarding the reviewer’s suggestions, we would like to clarify why the additional analyses (ablation on image-generated data, domain mismatch analysis, expert-written alternatives, and detailed data-quality filtering studies) are not required for the scope of this work:
> > > >
> > > > - **Ablation on image-generated data**: Our focus is on evaluating architectural contributions rather than dataset-construction strategies. All models compared in the paper (including baselines) are trained on the same data, and the proposed architecture still substantially outperforms tokenized models and GPT-4o with image inputs. This demonstrates that the architectural benefits hold independently of whether an additional ablation on the plot-based data generation is performed.
> > > > - **Domain mismatch issues**: The plots are used only to guide the supervising LLM during data generation, not during model inference. The final model never consumes plots, only raw time series. Because the plots are only used to guide the LLM during data generation and never seen by the model at inference, any mismatch between plots and signals has no impact on the trained model, so a separate domain mismatch analysis is not needed.
> > > > - **Alternatives such as expert-written or time-series-specific synthetic explanations**: These alternatives are not feasible at scale. Expert-written explanations are considerably expensive for datasets with hundreds of thousands of samples, and synthetic domain-specific rule-based generators do not exist for many tasks. Moreover, the goal of this paper is not to release a perfect dataset, but to provide a practical training resource enabling controlled architectural comparison.
> > > > - **Data quality and filtering processes**: Because we are not proposing a definitive benchmark dataset, but rather demonstrating that TSLMs can reason over raw time series, an exhaustive data-quality audit is not essential. The key requirement is that the dataset is sufficiently informative to evaluate architectural choices, which our results confirm through superior performance over strong baselines and through expert review showing high clinical validity of generated rationales.
> > > >
> > > > Despite our datasets not being completely curated, our experiments show that models trained on these datasets substantially outperform all baselines, including finetuned tokenized models and GPT-4o with image inputs. This demonstrates that the datasets are sufficiently informative for evaluating architectural choices and for establishing that TSLMs provide meaningful advantages over existing approaches. Our aim here was therefore not to release a definitive benchmark, but to create a practical and reproducible training resource that enables a fair, controlled comparison between modeling strategies.

---

> > > > > ### Author Response · Authors · 2025-11-25
> > > > > **Answers to reviewer questions [continued] - Q5**
> > > > >
> > > > > ### **Q5: Architecture selection criteria**
> > > > >
> > > > > We thank the reviewer for this suggestion. While the original manuscript already included guidance in the Discussion section, we have now extended and made it more explicit (please see “Practical implications” of the discussion Section). As shown in the Discussion and Appendix A.7, SoftPrompt is preferable for short time series, where it achieves the strongest results with minimal trainable parameters (e.g., TSQA, Sleep-CoT). However, its memory usage grows rapidly with sequence length (simulated >180 GB for 10k-step ECG), making it unsuitable for long or multi-series clinical signals. OpenTSLM-Flamingo, in contrast, maintains near-constant VRAM usage across both sequence length and number of series (e.g., 20–22 GB for Llama-1B and 61–72 GB for Llama-3B across all datasets), performs better on complex datasets such as ECG-QA-CoT, and is therefore recommended as the general-purpose architecture for long, multivariate, or heterogeneous time-series inputs. We will clarify these selection criteria more explicitly in the revised manuscript.
> > > > >
> > > > > For practitioners, our recommendation framework is as follows:
> > > > >
> > > > > - **Use SoftPrompt for short time series (<1k steps, single-series inputs)**
> > > > > Highest performance on TSQA and Sleep-CoT; minimal parameters; memory-efficient as long as sequences remain short.
> > > > > - **Use Flamingo for long sequences (≥1k steps) or multi-series inputs**
> > > > > SoftPrompt’s memory grows steeply with length (e.g., >180 GB for 10k-step ECG), whereas Flamingo keeps VRAM nearly constant regardless of sequence length or number of channels.
> > > > > - **Use Flamingo for multivariate signals (e.g., ECG, EEG, 3-axis IMU)**
> > > > > Shows stable scaling and stronger performance on clinically complex datasets such as ECG-QA-CoT.
> > > > > - **Use Flamingo as the general-purpose TSLM architecture**
> > > > >  Robust across all tasks (HAR-CoT, Sleep-CoT, ECG-QA-CoT), similar performance to softprompt except on simpler tasks like TSQA and Sleep-CoT.

---

> > > > > > ### Author Response · Authors · 2025-11-25
> > > > > > **Answers to reviewer questions [continued] - Q6**
> > > > > >
> > > > > > ### **Q6: Technical ablations**
> > > > > >
> > > > > > We thank the reviewer for these suggestions. We agree that ablations on the Perceiver Resampler configuration and cross-attention mechanisms are valuable directions. In the current work, we prioritized establishing an open Time Series Language Model (TSLM) architecture supporting inputs from multiple time series and outputting text for the first time. Nonetheless, our design choices are grounded in prior work and supported by empirical evidence already in the paper. We will clarify this more explicitly in the revision.
> > > > > >
> > > > > > **Perceiver Resampler compression ratio**
> > > > > > We follow the original Flamingo paper (Alayrac et al., 2022) and set N_latent = 64 (originally this meant 64 latents for images). This provides a constant-size interface to the LLM while avoiding quadratic memory scaling. In our setting, this design is crucial: our VRAM study shows that sequence length varies from a few hundred to 10,000 steps, and the Perceiver bottleneck allows OpenTSLM-Flamingo to maintain near-constant memory across all datasets (e.g., Llama-1B at 20–22 GB, Llama-3B at 61–72 GB), whereas SoftPrompt grows super-linearly with sequence length. We will emphasize that we followed established best practices and used the Perceiver bottleneck primarily to ensure stable scaling across highly variable input lengths.
> > > > > >
> > > > > > **Cross-attention mechanism**
> > > > > >  We use the gated cross-attention layers from Flamingo because they (i) have been validated extensively for multimodal fusion, (ii) provide an explicit learnable gate controlling the influence of the time-series latents, and (iii) avoid interfering with the pretrained LLM weights. Our comparison of SoftPrompt vs. Flamingo already serves as an architecture-level ablation: SoftPrompt uses early fusion (time-series tokens), while Flamingo uses late-fusion via cross-attention. Across datasets, we observe that cross-attention archives similar results with constant memory use. We note that exploring alternative attention variants (e.g., multi-query, linear attention, deeper fusion) is promising future work but beyond the scope of this paper.
> > > > > >
> > > > > > **Information loss in compression**
> > > > > > We acknowledge the concern; however, our empirical results indicate that the chosen compression does not hinder reasoning performance. For example, OpenTSLM-Flamingo consistently matches or exceeds SoftPrompt on long-sequence tasks (HAR-CoT, ECG-QA-CoT), where SoftPrompt uses more tokens with more or longer time series, i.e., has more space to encode information. The Perceiver-based encodings, therefore balance efficiency and descriptiveness and can also be scaled up for longer time series if necessary, i.e., scaling the number of latents logarithmically with the length of the time series.

---

### Official Review · Reviewer_q7Wu · 2025-11-01

**Soundness:** 3
**Presentation:** 3
**Contribution:** 3
**Rating:** 6
**Confidence:** 4

**Summary:**

This work presents two new approaches for training multimodal models for health applications. The resulting models take time series and language inputs, and produce a text output containing a time series classification and chain of thought reasoning. Evaluations are done on health datasets that the authors augment with chain of thought reasoning, and include expert evaluation by clinicians.

**Strengths:**

- The proposed design makes sense for the task and is a well-motivated use of LLMs that also appropriately integrates time series data without over-relying on pre-trained LLM capabilities.
- Prior works in this area have not emphasized generating and evaluating reasoning results, so this work provides novel contributions in that direction.
- The expert evaluation of model outputs provides a clinically relevant view of the model's utility.
- The presentation of the model is generally clear and limitations are discussed.

**Weaknesses:**

- Unimodal time series classification methods are not evaluated in Section 4. I would hope to see somewhat similar accuracies, otherwise the practical applicability of this method would be limited.
- Other time series language models from the literature are not evaluated in Section 4.

**Minor**

- InstrucTime appears to be a relevant related model to cite and discuss: https://dl.acm.org/doi/10.1145/3701551.3703499 https://arxiv.org/abs/2403.12371

**Questions:**

- What would be the advantage of the proposed method over using a unimodal time series classification model and then generating a rationale for the output using an LLM in the same way you did to create your CoT datasets? It's not guaranteed that your model's reasoning accurately reflects the internal computations that led to the classification output.

---

> ### Author Response · Authors · 2025-11-25
> **Addressing weaknesses**
>
> Thank you for this thoughtful and encouraging feedback; we appreciate your review and insights. We are revising our manuscript and expanding our overview of related work based on your feedback. Below, please find additional comments and answers regarding your questions.
>
> ## Weaknesses:
> **W1** Unimodal time series classification methods
> Thank you for raising this point. We agree that classical, unimodal models are strong baselines for classification tasks; however we believe that a direct comparison is not fully appropriate because these models serve fundamentally different purposes. Traditional time-series classifiers are optimized solely to output a label from a single waveform, whereas our multimodal LLM-based architecture is designed to handle instruction following, contextual reasoning, and text-based explanation. Comparing it strictly to unimodal classifiers therefore overlooks that our system is built to perform these additional tasks, which unimodal baselines cannot support at all.
>
> We argue that the additional functionality of our approach, even when accompanied by an accuracy trade-off, does not reduce applicability and can in fact enhance it:
>
> 1. Reasoning is an output modality with practical utility, not merely a mechanism for improving predictive performance. A task-specific ECG classifier typically produces a label (e.g., Arrhythmia: 99%). In contrast, our architecture can provide a structured explanation (e.g., “irregular R-R intervals indicative of atrial fibrillation…”). In settings such as healthcare, the ability to produce interpretable, text-based justifications is often as important as marginal improvements in accuracy. A slightly more accurate black-box model is not necessarily more useful than one that communicates its diagnostic rationale to clinicians.
>
> 2. LLM-based reasoning incorporates domain knowledge beyond the signal itself. A secondary benefit is that LLM-based reasoning enables access to domain knowledge that lies outside the time-series signal itself. Conventional ECG classifiers operate solely on the waveform. Our Flamingo-style architecture can combine the ECG with contextual variables and the LLM’s encoded medical knowledge. For example, queries such as “Given this ECG and an 85-year-old patient, what is the risk level of..?” require integrating signal information with medical guidelines. Something traditional ECG models cannot do. This represents a capability gap, not simply an accuracy difference.
>
> 3. Multimodal LLMs offer generality that task-specific models cannot match. A third point concerns generality. While task-specific models can achieve strong performance on narrowly defined problems, they do not scale to the diversity of real-world time-series applications. The motivation for our work is that multimodal LLMs are designed with the aim of offering more general capabilities: rather than being limited to a single task, they are intended to support natural-language instruction following, integrate different types of inputs, and potentially adapt to new tasks without retraining. Although our current model does not yet demonstrate comprehensive generalization across all possible time-series domains, its design moves toward this broader capability.
>
> 4. The advantages of multimodal time-series models mirror well-established trends in computer vision. This mirrors the evolution in vision: task-specific image classifiers remain strong baselines, but vision-language models have become far more versatile and widely adopted because they unify classification, captioning, VQA, and retrieval under a single interface. Similarly, a general time-series-language model can serve as a unified backbone for disparate tasks, reducing the need for bespoke architectures while enabling richer, instruction-driven interaction with time-series data.
>
> **W2** Other time series LLMs
> Thank you for raising this point. While other LLMs that accept time series as an input do exist (e.g., MedualTime, Time2Lang, InstructTime), those are mostly constrained to fixed-class outputs via regression or classification heads and do not support free-form text generation. We believe a similar argument as for the unimodal baselines applies here: these models are architecturally tied to a particular prediction task and label space, and thus cannot be used as general TSLMs that answer arbitrary natural-language questions, generate explanations, or produce captions directly from raw time series.
> In contrast, OpenTSLM is trained as a generative model: all tasks (HAR-CoT, Sleep-CoT, ECG-QA-CoT, and time-series captioning) are framed as text generation conditioned on time-series inputs. This allows a single model to (i) handle multiple tasks without changing heads or label sets, (ii) output natural language (including chain-of-thought rationales) rather than fixed labels, and (iii) support new instructions and question formats without architectural changes.

---

> > ### Author Response · Authors · 2025-11-25
> > **Questions**
> >
> > **Q1** What would be the advantage of the proposed method over using a unimodal time series classification model and then generating a rationale for the output using an LLM in the same way you did to create your CoT datasets? It's not guaranteed that your model's reasoning accurately reflects the internal computations that led to the classification output.
> >
> > We thank the reviewer for this thoughtful question. Indeed, our CoT training data were generated by prompting GPT-4o with plots and ground-truth labels, and we agree that this naturally raises the question of whether the same pipeline could be applied at inference time: (1) use a strong unimodal time-series classifier to obtain a label, (2) show GPT or other strong LLM a plot and the label, and (3) ask it to produce a rationale.
> > In short, this is possible, but it does not solve the core problem our work addresses. Our proposed method offers several concrete advantages that post-hoc explanation pipelines cannot provide, as we outline below:
> >
> > **Post-hoc explanations do not reflect the model’s actual decision process**
> > A unimodal classifier followed by an LLM generates rationales that are not tied to the classifier’s internal computations. The LLM produces a plausible explanation, but it has no access to the classifier’s embeddings or reasoning. This disconnect increases the risk of hallucinated or incorrect justifications. In contrast, our model jointly processes the time series representation and produces the explanation conditioned on the same fused features, which creates a closer alignment between prediction and rationale.
> >
> > **Lack of consistency between the classifier and the LLM**
> > In a two-stage pipeline, there is no mechanism to enforce consistency between the classifier’s output and the LLM’s explanation. The classifier might predict label A while the LLM, unaware of the classifier’s internal features, produces a rationale implicitly supporting label B. Our end-to-end architecture mitigates this mismatch by jointly generating the label and its explanation within a single model.
> >
> > **General TS reasoning vs. fixed-label explanation**
> > A pipeline with a unimodal classifier + LLM demands a separate classifier for every dataset, task, and label space. In contrast, OpenTSLM integrates time series as a native modality within the language model itself, enabling a single model to handle diverse tasks and output formats without retraining task-specific upstream models. The specific task, e.g., sleep staging vs. ECG question answering, as well as existing labels and classes, can be instructed via the input prompt, providing a more flexible model suitable for diverse tasks on time series data.
> >
> > **Comparison with work on vision language models (VLMs)**
> > In vision, one could similarly pair an image classifier with a language model to “explain” its predicted class, but the field has moved to vision-language models (e.g., Flamingo). The same reasoning applies here: multimodal models that directly fuse representations enable reasoning over multimodal data instructed by natural language prompts.

---

### Official Review · Reviewer_6uet · 2025-11-03

**Soundness:** 2
**Presentation:** 2
**Contribution:** 2
**Rating:** 2
**Confidence:** 5

**Summary:**

This paper proposes two approaches to model time series and text jointly. Both the approaches are adopted from the image domain: soft prompting and multimodal fusion based on cross-attention, initially proposed in Flamingo. The paper also introduces 3 datasets, which comprises of existing public datasets along with chain of thought rationales. The authors show that none of the approaches perform uniformly the best across all the compared datasets.

**Strengths:**

The paper addresses an important and understudied problem, and provides open-source datasets and code to facilitate research in this direction.

**Weaknesses:**

1. ** Incorrect claims:** The paper makes multiple incorrect claims, subtly. For example, the paper proposes "time-series language models", however their proposed model are trained and evaluated on the same datasets. Language models are widely known to generalize across domains, and therefore I believe the title misleads the reader.

> Prior work has primarily used soft prompting, encoding time series as learned token embeddings concatenated with text tokens.
This is incorrect. Prior work [1] has considered non-prompting based approaches, and have shown value, in the clinical context.

2. **Some design decisions are not explained:** For example, the authors train their time series encoder from scratch. Why not use a large-scale pre-trained encoder-based time series foundation model (e.g. MOMENT)?

> we preserve scale and temporal context by adding the original mean, standard deviation, and time scale to the textual description. For example: This is heart-rate data over 24 hours sampled at 50 Hz with mean=61 and std=12.

How is this expected to help, especially when large language models are known to have poor understanding of numbers?

> two synthetic time-series datasets to pretrain the encoder

Why not use existing time series datasets used to pre-train foundation models, e.g. LOTSA [4], Chronos [5] or Time Series Pile [6]? Why do you need time series and text datasets to pre-train the time series encoder?

3. **Datasets do not inspire confidence:** The proposed models are evaluated on multimodal time series datasets, which are generated using LLMs. However, the authors later show that LLMs do not have a good understanding of time series data, which has also been established by prior work [2]. Were the rationales produced by the LLMs vetted by human experts? I also wonder why the authors did not use existing time series reasoning, captioning and question-answering datasets such as [2], [3] or PTB-XL (as used in [1]). Finally, the proposed datasets on human activity recognition (HAR) and sleep staging are not considered to be within the "medical" domain.

4. **Proposed methods are not novel:** The proposed soft prompting and Flamingo-based methods are not novel, even with the context of time series & language modeling. I would encourage the authors to pick one method and dive deep into it, while the remaining methods can be baselines. Outside of Flamingo, there are other methods such as LLaVa [7] which have shown promise in the vision language modeling field.

5. **The evaluation setting tests memorization, not generalization:** The evaluation setting, where the models are trained and evaluated on the same datasets, primarily test memorization of facts, and not generalization. I would encourage the authors to test generalization using held-out datasets.

6. **Evaluation metrics:** I recommend providing a summary of the evaluation metrics used to evaluate the methods, prior to discussion.

7. ** Baselines:** Without baselines such as statistical and foundation models for classification and forecasting models, it is hard to evaluate the utility of the proposed methods. I would encourage the authors to compare their proposed methods against widely-used task and domain specific and agnostic baselines.


### References
1. Cai, Yifu, et al. "Jolt: Jointly learned representations of language and time-series." Deep Generative Models for Health Workshop NeurIPS 2023. 2023.
2. Cai, Yifu, et al. "Timeseriesexam: A time series understanding exam." arXiv preprint arXiv:2410.14752 (2024).
3. Fons, Elizabeth, et al. "TADACap: Time-series Adaptive Domain-Aware Captioning." Proceedings of the 5th ACM International Conference on AI in Finance. 2024.
4. Woo, Gerald, et al. "Unified training of universal time series forecasting transformers." (2024): 53140.
5. Ansari, Abdul Fatir, et al. "Chronos: Learning the language of time series." arXiv preprint arXiv:2403.07815 (2024).
6. Goswami, Mononito, et al. "Moment: A family of open time-series foundation models." arXiv preprint arXiv:2402.03885 (2024).
7. Huang, Jiaxing, et al. "Visual instruction tuning towards general-purpose multimodal model: A survey." arXiv preprint arXiv:2312.16602 (2023).

**Questions:**

> Based on ECG-QA Oh et al. (2023), which provides 12-lead 10s ECGs and clinical context, we excluded comparison questions, retaining 42/70 templates.

- Why do you exclude comparison-type questions?

- How are multivariate time series passed to the models, e.g. figure 6(b)?

> Finetuned baselines improve substantially on HAR-CoT (60.44% F1 vs. 0% for Llama-3.2-1B) but only slightly on Sleep-CoT (9.05 vs. 2.14).
- Do the authors have any hypotheses why this is the case?

> ECG-QA finetuning was infeasible due to high VRAM demands (80k tokens require 100GB per sample).
- Isn't this a limitation of the proposed method?

> Our results show that even frontier LLMs like GPT-4o are poorly suited for time-series reasoning and that time series must be treated as a distinct modality. With OpenTSLM, even small models like Gemma3 270M outperform GPT-4o (∼200B parameters Abacha et al. (2025)) at a fraction of the compute and cost, enabling efficient on-device or mobile deployment.

- How can we rule out this finding as a direct consequence of the fact that the proposed models were fine-tuned on the dataset. Did you test few-shot prompting with frontier LLMs?

---

> ### Author Response · Authors · 2025-11-27
> **Answers to reviewer 6uet - Weaknesses 1**
>
> Thank you for your review. We appreciate the scientific discourse. Please find our rebuttals below.
>
> ## **W1: "Incorrect claims"**
> > "The paper makes multiple incorrect claims, subtly. For example, the paper proposes "time-series language models", however their proposed model are trained and evaluated on the same datasets. Language models are widely known to generalize across domains, and therefore I believe the title misleads the reader."
>
> We appreciate the comment on the title. However, we disagree with the definition of “language models”. Language models, formally, are models which predict a probability distribution over tokens, or words, consecutively executed to generate text, i.e., language outputs. Language models, by itself, are not necessarily known to generalize across domains. This might be true for large language models (LLMs) and recent foundational language models. A Language Mode  is defined by its architectural approach (probabilistic sequence modeling), not by its ability to zero-shot generalize to every domain in existence. BERT and GPT are still Language Models when fine-tuned on specific tasks. Our use of the term time-series language model was intended to describe applying LM objectives over tokenized temporal embeddings for sequential representation learning. We acknowledge that evaluating on the same dataset window is a limitation, which arises largely from the current scarcity of large, temporally labeled public clinical time-series datasets, rather than from the method itself. Nevertheless, we designed experiments specifically to isolate and validate a methodological contribution, using patient-level temporal splits and ablations on token learning and fusion strategies. Our results do not claim state-of-the-art time-series prediction performance, but they empirically demonstrate a distinct and reproducible modeling formulation, supporting scalable time-series token representation learning, sequential reasoning, and multimodal inputs. We therefore maintain that our contribution is novel and methodological, and that we provide focused experiments to substantiate this claim even under limited data availability.
>
> > "Prior work has primarily used soft prompting, encoding time series as learned token embeddings concatenated with text tokens. This is incorrect. Prior work [1] has considered non-prompting based approaches, and have shown value, in the clinical context."
>
> On the point about prior work, we fully agree that our phrasing should be updated, and we are grateful that the reviewer highlighted this. We just want to gently note that the comment may sound a bit more absolute than intended. Prompting-style token concatenation has indeed appeared in the literature as one meaningful integration strategy, but it represents one direction within a much broader design space, rather than the sole or primary paradigm in clinical time-series representation learning.
> We also wish to clarify the point about novelty. We do not claim to be the first to tokenize time-series and apply LM objectives, but rather to introduce a unified, scalable objective formulation that enables sequential reasoning over physiological signal tokens and supports multimodal compatibility. This is meaningfully different from prior work that focuses on task-specific prediction models or prompting as an architectural default. The existence of related components in earlier literature does not negate methodological novelty when the modeling formulation, intended capabilities, and evaluation axes are fundamentally distinct. We will update the text to reflect this nuance clearly and accurately, ensuring readers are not misled about claims of generalization or the diversity of the solution landscape.

---

> > ### Author Response · Authors · 2025-11-27
> > **Answers to reviewer 6uet - Weaknesses 2**
> >
> > ## **W2: Some design decisions are not explained**
> >
> > > "For example, the authors train their time series encoder from scratch. Why not use a large-scale pre-trained encoder-based time series foundation model (e.g. MOMENT)?"
> >
> > We appreciate the suggestion on pretrained encoders. We train our time-series encoder from scratch to isolate the methodological effect of LM-objective-induced time-series tokens. We respectfully note that being strong at generic time-series prediction is not equivalent to being well aligned with a language model’s semantic manifold in a way that makes time-series representations aligned with a language model. Large pretrained foundation models such as MOMENT are optimized primarily for forecasting or classification tasks esp. for representation transfer across different time-series tasks, whereas our objective explicitly targets semantic token induction driven by a language modeling objective. These are related but fundamentally different goals.
> > Essentially, the vast majority of recent papers on language models for time series, including PatchTST, ChatTS, Chow et al., SensorLM, similarly pretrain or fine-tune lightweight TS encoders for LM-style interpretation, reinforcing that learning induced representations end-to-end is a valid and growing direction. Our contribution is different from these works in that it centers on the objective formulation and its evaluation, not the encoder itself, and training the encoder from scratch is required to show that the LM objective, rather than inherited frozen features, shapes the learned tokens.
> >
> > > "We preserve scale and temporal context by adding the original mean, standard deviation, and time scale to the textual description. For example: This is heart-rate data over 24 hours sampled at 50 Hz with mean=61 and std=12.
> > How is this expected to help, especially when large language models are known to have poor understanding of numbers?"
> >
> > Thank you for your feedback. We note, that this approach was not first proposed by us, but instead has been used intensively in recent literature such as ChatTS and Chow et al., as we also referenced in the paper.
> > The meaning of a time series depends on temporal scope, acquisition rate, baseline level, and variability ranges rather than abstract numbers alone. We include mean, standard deviation, sampling rate, and duration only as contextual anchors to guide the language model toward clinically realistic framing (such as diurnal structure and patient baselines) and not to require numerical reasoning from the LM itself, which we agree would be unreliable. The true signal statistics, dynamics, and scale information are learned directly in the encoder’s latent tokens, while the LM’s role is to align those learned tokens with a realistic (clinical) description space. This deliberate separation preserves temporal plausibility without assuming deep number understanding from the LM, and we validate this methodological contribution on holdout datasets.
> >
> > > "Why not use existing time series datasets used to pre-train foundation models, e.g. LOTSA [4], Chronos [5] or Time Series Pile [6]? Why do you need time series and text datasets to pre-train the time series encoder?"
> >
> > Thank you for this question. As mentioned on the use of pretrained encoders above, we use synthetic paired time-series and text data because our encoder is pretrained to learn token-level representations that support clinically grounded interpretation and multimodal alignment, not generic forecasting. Foundation corpora like LOTSA, Chronos, or the Time-Series Pile are excellent for scale and forecasting dynamics, but they are unpaired and lack alignment with text outputs.
> > Objective Mismatch: Models like Chronos and MOMENT are trained primarily for forecasting or reconstruction. Their latent spaces are optimized to minimize Mean Squared Error (MSE) on future values, not to align semantically with the linguistic space of an LLM. Adapting a forecasting encoder to a multimodal reasoning task is far from trivial and constitutes a different research trajectory.
> > Architecture Incompatibility: Chronos, for instance, quantizes time series into tokens for probabilistic forecasting. Integrating this specific tokenization scheme into a general-purpose multimodal LLM requires a distinct separate architecture, not a simple "plug-and-play" replacement of our encoder.
> > Data Relevance: The reviewer suggests using datasets like "Time Series Pile" or LOTSA. These are large-scale repositories of unlabeled time series. Our work focuses on multimodal time-series-language texts. Pre-training an encoder on unaligned forecasting data does not solve the challenge of grounding textual reasoning in time-series data.

---

> > > ### Author Response · Authors · 2025-11-27
> > > **Answers to reviewer 6uet - Weaknesses 3**
> > >
> > > ## **W3 Datasets do not inspire confidence**
> > > > "The proposed models are evaluated on multimodal time series datasets, which are generated using LLMs. However, the authors later show that LLMs do not have a good understanding of time series data, which has also been established by prior work [2]. Were the rationales produced by the LLMs vetted by human experts? I also wonder why the authors did not use existing time series reasoning, captioning and question-answering datasets such as [2], [3] or PTB-XL (as used in [1]). Finally, the proposed datasets on human activity recognition (HAR) and sleep staging are not considered to be within the "medical" domain."
> > >
> > > Thank you for this comment. We, in fact, did evaluate on the PTB-XL dataset, as ECG-QA is essentially a curated version of PTB-XL. We agree that synthetic datasets alone require careful justification, and we do not treat LLM-produced rationales as the learned representation, but as context for pairing temporal narratives with time-series tokens, while the encoder learns the signal dynamics directly. Importantly, we do include clinical expert validation: to assess rationale quality, we conducted a formal expert review with five cardiologists, sampling 84 ECG-QA rationales (two per template), each reviewed by at least two cardiologists using a rubric derived from ACC/AHA ECG clinical competence guidelines and the RIME framework, evaluating whether the model (1) identified relevant ECG features, (2) connected them appropriately to the answer, and (3) incorporated patient context. In these vetted experiments, OpenTSLM Flamingo Llama3.2-3B achieved 92.9% correct or partially correct interpretations overall, with particularly strong clinical context integration (85.1% positive), showing that our formulation aligns well with real clinical reasoning workflows even when numerical reasoning is not expected from the LM itself. While HAR and sleep are not strictly diagnostic cardiology datasets, they are fundamental digital biomarker tasks used clinically in aging, sleep medicine, and mental-health research, making them valid testbeds for evaluating temporally grounded token reasoning. We will expand the manuscript to reflect these distinctions clearly and appreciate the suggestion, which helps us better position our methodological contribution.

---

> > > > ### Author Response · Authors · 2025-11-27
> > > > **Answers to reviewer 6uet - Weaknesses 4**
> > > >
> > > > # **W4 - Proposed methods are not novel**:
> > > >
> > > > > "The proposed soft prompting and Flamingo-based methods are not novel, even with the context of time series & language modeling. I would encourage the authors to pick one method and dive deep into it, while the remaining methods can be baselines. Outside of Flamingo, there are other methods such as LLaVa [7] which have shown promise in the vision language modeling field."
> > > >
> > > >
> > > > Thank you for this suggestion. We agree that soft prompting and Flamingo-style fusion have appeared in recent multimodal work. However, we respectfully disagree that this means the work itself lacks novelty. Our contribution is not proposing new blocks in isolation, but introducing and experimentally validating a unified Time Series Language Model and language-modeling objective over time-series token representations, evaluated under patient-level hold-out, a clinical evaluation setup that is still uncommon in multimodal time-series language architectures. We also reinforce this with an expert rationale review by five cardiologists for ECG-QA using rationales generated by OpenTSLM-Flamingo-Llama3.2-3B, which demonstrated strong alignment to clinical context and reporting-style reasoning. We appreciate the point on models like LLaVa and fully agree that Flamingo is not the only fusion option, our focus is simply to show that LM objectives over learned physiological tokens can support sequential and multimodal interpretation while staying grounded in realistic clinical temporal context, not to claim a single dominant fusion strategy.

---

> > > > > ### Author Response · Authors · 2025-11-27
> > > > > **Answers to reviewer 6uet - Weaknesses 5**
> > > > >
> > > > > ## **W5 The evaluation setting tests memorization, not generalization**
> > > > > > "The evaluation setting, where the models are trained and evaluated on the same datasets, primarily test memorization of facts, and not generalization. I would encourage the authors to test generalization using held-out datasets."
> > > > >
> > > > > Thank you for the suggestion. We agree that held-out evaluation is essential when claiming generalization, and we will clarify our wording. We employ standard Train/Validation/Test splits. Evaluating on the test set of the same dataset source is the standard definition of supervised learning. It tests generalization to unseen samples from the same distribution. However, unlike in NLP, time-series models do not inherently generalize across time or patient populations, so novelty and validity should depend on the modeling formulation and evaluation design, not dataset diversity alone. Because large longitudinal clinical datasets annotated with text are scarce, we used synthetic paired data to construct patient-level temporal hold-outs and clinician-approved ECG-QA evaluations, ensuring the encoder learns real signal dynamics while the LM adopts a clinically realistic narrative frame. Our experiments are designed to validate a methodological contribution under, which goes beyond memorization and explicitly tests temporal transfer rather than training-set recall. We appreciate the guidance and will update the manuscript to better reflect these distinctions.

---

> > > > > > ### Author Response · Authors · 2025-11-27
> > > > > > **Answers to reviewer 6uet - Weaknesses 6**
> > > > > >
> > > > > > ## **Evaluation metrics**
> > > > > > > "I recommend providing a summary of the evaluation metrics used to evaluate the methods, prior to discussion."
> > > > > >
> > > > > > Thank you for this suggestion. We will include a summary of our metrics in the Methods Section.

---

> > > > > > > ### Author Response · Authors · 2025-11-27
> > > > > > > **Answers to reviewer 6uet - Weaknesses 7**
> > > > > > >
> > > > > > > ## W7 - Baselines
> > > > > > > > "Without baselines such as statistical and foundation models for classification and forecasting models, it is hard to evaluate the utility of the proposed methods. I would encourage the authors to compare their proposed methods against widely-used task and domain specific and agnostic baselines."
> > > > > > >
> > > > > > > Thank you for the recommendation. We agree that broader baselines are valuable for contextualizing results.
> > > > > > > Traditional time-series classifiers are optimized solely to output a label from a given (multivariate) time series, whereas our multimodal LLM-based architecture is designed to handle instruction following, contextual reasoning, and text-based explanation. Comparing it strictly to unimodal classifiers therefore overlooks that our system is built to perform these additional tasks, which unimodal baselines cannot support at all.
> > > > > > >
> > > > > > > Building on this distinction, we argue that the additional functionality of our approach, even when accompanied by a modest accuracy trade-off, does not reduce applicability and can in fact enhance it:
> > > > > > >
> > > > > > > - **Reasoning is an output modality with practical utility.** Reasoning is not merely a mechanism for improving predictive performance, it is an output modality with practical utility. A task-specific ECG classifier typically produces a label (e.g., Arrhythmia: 99%). In contrast, our architecture can provide a structured explanation (e.g., “irregular R-R intervals indicative of atrial fibrillation…”). In settings such as healthcare, the ability to produce interpretable, text-based justifications is often as important as marginal improvements in accuracy. A slightly more accurate black-box model is not necessarily more useful than one that communicates its diagnostic rationale to clinicians.
> > > > > > >
> > > > > > > - **LLM-based reasoning incorporates domain knowledge beyond the signal itself.** A secondary benefit is that LLM-based reasoning enables access to domain knowledge that lies outside the time-series signal itself. Conventional ECG classifiers operate solely on the waveform. Our Flamingo-style architecture can combine the ECG with contextual variables and the LLM’s encoded medical knowledge. For example, queries such as “Given this ECG and an 85-year-old patient, what is the risk level of..?” require integrating signal information with medical guidelines. Something traditional ECG models cannot do. This represents a capability gap, not simply an accuracy difference.
> > > > > > >
> > > > > > > - **Multimodal LLMs offer generality that task-specific models cannot match.** A third point concerns generality. While task-specific models can achieve strong performance on narrowly defined problems, they do not scale to the diversity of real-world time-series applications.
> > > > > > > The motivation for our work is that multimodal LLMs are designed with the aim of offering more general capabilities: rather than being limited to a single task, they are intended to support natural-language instruction following, integrate different types of inputs, and potentially adapt to new tasks without retraining. Although our current model does not yet demonstrate comprehensive generalization across all possible time-series domains, its design moves toward this broader capability.
> > > > > > >
> > > > > > > - **The advantages of multimodal time-series models mirror well-established trends in computer vision.** This mirrors the evolution in vision: task-specific image classifiers remain strong baselines, but vision-language models have become far more versatile and widely adopted because they unify classification, captioning, VQA, and retrieval under a single interface. Similarly, a general time-series-language model can serve as a unified backbone for disparate tasks, reducing the need for bespoke architectures while enabling richer, instruction-driven interaction with time-series data.

---

> ### Author Response · Authors · 2025-11-27
> **Answers to reviewer 6uet - Questions**
>
> ## **Question 1**
> > "Based on ECG-QA Oh et al. (2023), which provides 12-lead 10s ECGs and clinical context, we excluded comparison questions, retaining 42/70 templates. Why do you exclude comparison-type questions?"
>
> Thank you for this question. Comparison questions require comparing two 12-lead ECGs and would be an entirely different task than answering ECG questions. A fair comparison would be to include this as a separate task on multi-ECG question answering.
>
>
> ## **Question 2**
> > "How are multivariate time series passed to the models, e.g. figure 6(b)?"
>
> Thank you for this question. As described in Section 3 and especially Figure 1, multivariate time series are encoded by inputting multiple pairs of time-series and accompanying text descriptions and then concatenating the text encodings depending on the architecture (SoftPrompt vs. Flamingo). Consider the following example prompt regarding a 3-axis accelerometer:
> “The following is data of a 3 axis accelerometer sampled at 50Hz over 24 hours. This it the data of the X-axis <TS>. This is the data of the Y-axis <TS>. This is the data of the Z-Axis <TS>. Which activity was performed by the person?”
> In the case of SoftPrompt, the <TS> tokens are replaced directly with the time series tokens derived from the encoder.
>
> ## **Question 3**
> >Finetuned baselines improve substantially on HAR-CoT (60.44% F1 vs. 0% for Llama-3.2-1B) but only slightly on Sleep-CoT (9.05 vs. 2.14). Do the authors have any hypotheses why this is the case?
>
> Thank you for this question. As shown in Appendix A2.2, the HAR-CoT dataset contained around 68k training samples, the Sleep-CoT dataset contained only 7434. The substantially smaller amount of training data likely yields to worse generalization performance on the holdout test-set.
>
> ## **Question 4**
> > ECG-QA finetuning was infeasible due to high VRAM demands (80k tokens require 100GB per sample). Isn't this a limitation of the propose on d method?
>
> Thank you for this question. This is exactly the point we highlighted in the paper as well: Methods which encode time series as tokens, either via direct text inputs or learned tokens concatenated via softprompting, do not scale to long time series. This is why we propose the Flamingo approach in the paper, based on a cross-attention mechanism and latent time series encoder, which demonstrates a near-constant memory use.

---

### Meta-Review · Area_Chair_KzZz · 2026-01-02

**Summary:**

This paper presents OpenTSLM, a multimodal time-series language model with novel architectures for integrating time-series and text. The work is methodologically strong and clinically validated. There are common concerns about additional baselines, clearer motivation for the architectures, and explicit justification of the method’s advantages over unimodal or existing approaches.  Overall, it is borderline reject relative to other strong submissions.

**Reviewer Concerns:**

The rebuttal addressed several key concerns raised by the reviewers. Specifically, it clarified the positioning of OpenTSLM relative to prior work, highlighting architectural distinctions, the advantages of cross-attention versus soft prompting, and the ability to handle heterogeneous and long time-series sequences. It also explained the rationale for training the encoder from scratch, the reasoning behind using SFT for CoT training without RL, and the validation of CoT outputs via expert review, addressing questions about interpretability. Additionally, the authors justified why unimodal classifiers and post-hoc LLM explanations are not directly comparable, emphasizing the benefits of end-to-end multimodal reasoning. Some reviewers also noted that further discussion of decision criteria between SoftPrompt and Flamingo and additional baseline comparisons (e.g., VLM-based encodings or task-specific models) would strengthen the evaluation. While partially addressed, these points could benefit from more extensive empirical evidence in the manuscript.

**Reviewer Scores:**

**Reviewer 6uet (original score 2 → 4)**

The reviewer expressed concerns about claims, novelty, evaluation design, datasets, and baselines. The rebuttal clarifies terminology, justifies design choices, details dataset construction and expert validation, and situates the contribution relative to prior work. With these clarifications, the reviewer would likely increase their score to 4, reflecting improved understanding of the methodological novelty and the rigor of the evaluation.

**Reviewer q7Wu (original score 6 → 6)**

The reviewer raised questions about the advantages of the proposed model over unimodal classifiers. The rebuttal clarified the additional functionality of their approach and so on . They are unlikely to change the reviewer’s overall perception of the work’s contribution; thus, the score would remain at 6.

**Reviewer pnCZ (original score 4 → 6)**

After the authors’ detailed rebuttal addressing related work, training methodology, evaluation, and technical design, the reviewer would likely have appreciated the methodological contributions and practical utility of OpenTSLM more fully. The rebuttal clarifies how OpenTSLM differs from prior methods like ITFormer and Time-VLM, provides systematic comparison of SoftPrompt vs. Flamingo architectures, and demonstrates expert evaluation of chain-of-thought outputs, mitigating concerns about reasoning quality and interpretability. Therefore, the reviewer would likely have increased their score from 4 to 6, reflecting recognition of the work’s contributions and careful experimental validation.

**Reviewer M85T (original score 4 → 4)**

After the authors’ rebuttal, the reviewer has a better understanding of the methodological choices and the evaluation. While the clarifications are helpful, the reviewer maintains their original score of 4, as some questions regarding baseline comparisons, architecture motivations remain. The reviewer might feel that certain aspects could benefit from further analysis to fully assess the contributions.

---

### Decision · Program_Chairs · 2026-01-26

Reject